# CLIMADAT-GRid: A high-resolution daily gridded precipitation and temperature dataset for Greece

Konstantinos V. Varotsos[1], George Katavoutas[1], Gianna Kitsara[1], Anna Karali[1], Ioannis Lemesios[1], Platon Patlakas[2], Maria Hatzaki[3], Vassilis Tenentes[1], Athanasios Sarantopoulos[4], Basil Psiloglou[1], Aristeidis G. Koutroulis[5], Manolis G. Grillakis[5], Christos Giannakopoulos[1]

[1]Institute for Environmental Research and Sustainable Development, National Observatory of Athens, Athens, 15236, Greece
[2]Department of Physics, National and Kapodistrian University of Athens, Athens, 15784, Greece
[3]Laboratory of Climatology and Atmospheric Environment, Section of Geography and Climatology, Department of Geology and Geoenvironment, National and Kapodistrian University of Athens, Athens, 15784, Greece
[4]Hellenic National Meteorological Service, Athens, 16777, Greece
[5]School of Chemical and Environmental Engineering, Technical University of Crete, Chania, 73100, Greece

*Correspondence to*: Konstantinos V. Varotsos (varotsos@noa.gr)

**Abstract.** We introduce the development of CLIMADAT-GRid, the first publicly available daily air temperature and precipitation gridded climate dataset for Greece at a high resolution of 1 km x 1 km, covering the period 1981–2019. The dataset is derived from quality-controlled and homogenized daily measurements from an extensive network of meteorological stations: 122 for temperature and 312 for precipitation. Several approaches are evaluated for generating daily gridded datasets, including Fixed Random Kriging, Generalized Additive Models, K-Nearest Neighbors, and Support Vector Machines. Based on the evaluation analysis against withheld observational data, Fixed Random Kriging is selected as the method for the CLIMADAT-GRid construction. To address the lack of a dense temperature observational network, high-resolution simulations from the WRF model are blended with observational data to produce the gridded temperature datasets. CLIMADAT-GRid is benchmarked against the CHELSA-W5E5, a global climate product with a similar resolution, for the overlapping period 1981–2016. While both datasets show comparable results for temperature, CLIMADAT-GRid demonstrates superior spatial performance and closer agreement with observational data for both the mean and the extreme values. Regarding precipitation, CLIMADAT-GRid consistency indicates higher values than CHELSA-W5E5, especially during the rainy season, but exhibits better agreement with observations. In terms of the number of wet days, both datasets overestimate spatial means relative to observations, with CLIMADAT-GRid showing a more pronounced orographic pattern than CHELSA-W5E5. Both datasets show similar results for the number of days with precipitation amounts equal to or higher than 10 mm, with CLIMADAT-GRid indicating better overall agreement with the observations. The CLIMADAT-GRid dataset is publicly available at https://doi.org/10.5281/zenodo.14637536 and can be cited as Varotsos et al. (2025).

## 1 Introduction

High-resolution gridded climate datasets, both spatially and temporally, are a valuable resource for research and information in climate studies as well as in other areas such as hydrology, agriculture, energy and health (Herrera et al., 2012). In addition, high-resolution gridded datasets are used to evaluate, bias adjust and statistically downscale both regional and global climate models and seasonal forecasts (Lorenz et al., 2021; Nilsen et al., 2022; Varotsos et al., 2023a; Karali et al., 2023). Depending on the data sources and derivation techniques, gridded climate datasets can be divided into two main categories: (i) reanalysis datasets and (ii) gridded observational datasets. Reanalysis datasets provide a numerical description of the recent climate by combining dynamical models that assimilate observations, while gridded observational datasets are based on the statistical transformation of point meteorological station data into grid using geostatistical modelling.

As for the second category, which is the focus of this study, the remarkable advances in computing power and software have led to the development and creation of gridded observational datasets at both global, regional/national and sub-national levels. These datasets include E-OBS (Cornes et al., 2018) which is the state-of-art daily gridded observational dataset for the entire European domain with a resolution of 0.1°, while on the regional/national and sub-national scale a number of datasets have recently emerged in Europe. These include Iberia01 (Herrera et al., 2019) for the Iberian Peninsula (daily gridded dataset for temperatures and precipitation at 0.1° grid), SPREAD (Serrano-Notivoli et al., 2017) and STEAD (Serrano-Notivoli et.al., 2019) for Spain (daily datasets for precipitation and temperatures at 5 km x 5 km, respectively), SiCLIMA (Serrano-Notivoli et al., 2024) for Aragon, Spain (daily dataset for precipitation and temperatures at 500 m x 500 m), PTHRES (Fonseca and Santos 2018) for Portugal (daily dataset for temperatures at 1 km x 1 km), HYRAS (Krähenmann et al., 2018) for Germany (hourly dataset for a number of variables at 1km x1km), HadUK-Grid (Hollis et al. 2019) for the United Kingdom (daily dataset for a number of variables at 1 km x 1 km), seNorge2 (Lussana et al., 2018a,b) for Norway (daily dataset for precipitation and temperatures at 1 km x 1 km, respectively), SLOCLIM (Škrk et al., 2021) for Slovenia (daily dataset for precipitation and temperatures at 1 km x 1 km), MeteoSerbia1km (Sekulić et al. 2021) for Serbia (daily dataset for a number of variables at 1 km x 1 km) and GAA.HRES (Varotsos et al., 2023a) for Attica, Greece (daily dataset for precipitation and temperatures at 1 km x 1 km). It is important for users to recognize that these gridded observational products are geostatistically generated, rather than direct observations. Consequently, they are subject to several limitations and the accuracy of these datasets largely depends on the quality and spatial density of the underlying meteorological station network. In particular, interpolation methods tend to perform poorly in regions with sparse station coverage or complex topography (Hofstra et al., 2010; Beguería et al., 2016; Herrera et al., 2019). While most of these datasets are built upon dense networks of ground-based observations, in areas with limited station density or insufficient representation of elevation gradients it is often required enhancement through the integration of satellite data, reanalysis products, and atmospheric models to improve spatial coverage and reliability (Doblas-Reyes et al., 2021; Varotsos et al., 2023a). It should be noted that Serrano-Notivoli and Tejedor (2021) analyzing the performance of 48 gridded products proposed a general workflow to

transform observations into grid estimates, which includes four steps: i) quality control, ii) data series reconstruction, ii) gridding and iv) assessment of the uncertainty.

Various gridding techniques for creating daily gridded datasets have been discussed in the existing literature. For instance, in the earlier versions of E-OBS (prior to v16, Haylock et al., 2008) and Iberia01 (Herrera et al., 2019), daily gridded datasets for temperatures (daily maximum, minimum and mean) and precipitation, were constructed using a trivariate thin-plate spline (using elevation as a covariate, Hutchinson et al., 2009) to construct monthly background field values (mean for temperatures and sums for precipitation), while the daily anomalies or proportions for temperatures and precipitation,

respectively were interpolated using ordinary Kriging. To obtain the final daily gridded datasets the aforementioned fields were superimposed by addition and multiplication for temperatures and precipitation, respectively. In the latter versions of E-OBS the daily gridded datasets for temperatures and precipitation were constructed using Generalized Additive Models (GAMs, Wood, 2017) to estimate the long-range spatial trend in the data, while Gaussian Random Field simulation was used to interpolate the GAMs residuals. Other approaches include multiple linear regression, Delaunay triangulation and optimal

interpolation (Nordic gridded temperature and precipitation data, NDCG, Tveito et al., 2000; Tveito et al., 2005; Lussana et al., 2018a, b). Furthermore, while machine learning (ML) has been successfully used to statistically downscale ERA5 (Qin et al., 2022; Hu et al., 2023) and climate change projections (Hernanz et al., 2024 and references therein), few studies, to our knowledge, have explored or evaluated the use of machine learning algorithms for generating observational gridded datasets. In particular, MeteoSerbia1km is a 1km horizontal daily gridded dataset for temperatures, mean sea-level pressure, and total

precipitation for the years 2000–2019, which was produced using the RFSI method, a spatial interpolation method based on the random forest ML algorithm (Sekulic et al., 2021). Moreover, Bonsoms and Ninyerola (2024) evaluated five ML techniques for the spatial interpolation of annual precipitation, minimum and maximum temperatures in the Pyrenees. The accuracy and performance of K-Nearest Neighbors, Supported Vector Machines, Neural Networks, Stochastic Gradient Boosting, and Random Forest were compared with those of multiple linear regressions and generalized additive models.

According to the authors regardless of the elevation range, the geographical sector under analysis, or the predictor variables used, the ML algorithms outperformed multiple linear regressions and generalized additive models. Nevertheless, the authors did not proceed to the construction of a gridded dataset based on ML techniques. This is most likely due to the nature of ML techniques, which were designed for feature space qualities that cover almost all types of data and are therefore not commonly employed for spatial modeling (Nwaila et al., 2024).

In this study, we introduce CLIMADAT-GRid: a high-resolution (1 km x 1 km) daily gridded dataset for air temperatures (daily maximum, minimum and mean) and precipitation for Greece, covering the period 1981–2019 (Varotsos et al. 2025). To the best of our knowledge, this is the first publicly available dataset for Greece that offers daily gridded temperatures and precipitation in Greece at such fine spatial resolution. Previous studies known to the authors have primarily focused on monthly values of these variables for the 1971–2000 period (Mamara et al., 2017; Gofa et al., 2019).

 **2 Data**

In this section, the datasets utilized in the analysis are presented. Subsection 2.1 summarizes the daily observational data, including maximum ($TX$), minimum ($TN$), and average ($TG$) temperatures, as well as daily precipitation ($PR$). Subsection 2.2 outlines the procedures applied for quality control, gap filling, and homogenization of the datasets. Subsection 2.3 describes the Weather Research and Forecasting (WRF) model simulation, whose output is blended with the available temperature 100 observational data using gridding techniques, as detailed in Sect. 3. This approach was preferred over relying solely on observational data due to the sparse spatial coverage of in situ measurements, especially at higher altitudes (above 1000 m) as presented in Subsect. 2.1.

**2.1 Daily observational data for maximum ($TX$), minimum ($TN$), mean ($TG$) temperatures and daily precipitation ($PR$)**

This study utilizes daily air temperature observations from two main sources. The first is the National Observatory of Athens Automatic Network (NOAAN, Lagouvardos et al., 2017), which provides records from 48 stations for the period 2010– 2019, and the second source is Hellenic National Meteorological Service (HNMS), which provides temperature records from 73 stations spanning 1981–2019. In addition, we incorporate daily observations from the historical weather station of the National Observatory of Athens in Thissio (NOA, Founda et al., 2022) for the same period. In total, daily data from 122 110 meteorological stations across Greece were collected (Fig. 1a), with station altitude ranging from 1 to 960 m above sea level (a.s.l.). Temperature data were aggregated over a 24-hour period from 00:00 to 24:00 UTC.

In addition to the data from the stations mentioned above, we also collected daily precipitation data for 190 stations provided by the General Secretariat for Natural Environment and Water of the Ministry of Environment and Energy for the period 1981–2019. In total, daily precipitation from 312 stations are obtained (Fig. 1b), with altitudes from sea level to 1130 m a.s.l. 115 Only stations with less than 10% missing data annually were considered. According to the data providers, daily precipitation data were collected over a 24-period from 08:00 to 08:00 UTC for the HNMS, NOA and the stations provided by the General Secretariat for Natural Environment and Water of the Ministry of Environment and Energy. Regarding the NOAAN stations, daily precipitation data were collected over a 24-period from 00:00 to 24:00 UTC.


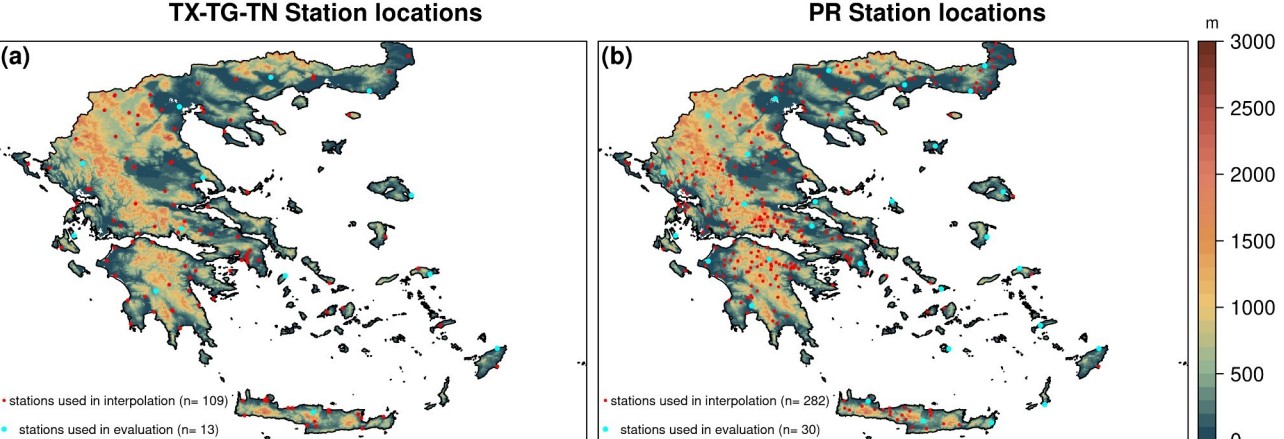

**Figure 1. Locations of meteorological stations used for (a) temperatures and (b) precipitation measurements, including both the stations used in the interpolation and the withheld stations used for evaluation. The background shows elevation data from the Global Multi-resolution Terrain Elevation Data 2010 (GMTED2010).**

## 2.2 Quality control, gap filling and homogenization

An initial quality control for all variables was conducted using the R package climatol (version 4.1.1, Guijarro, 2023), which automatically identifies and discards all extreme anomalies and removes prolonged sequences of identical values from data. As for daily precipitation, which has a significantly skewed frequency distribution, the deletion of high isolated data is not permitted because heavy rain can occur between two days with little or no precipitation. In addition, zero values of daily precipitation are automatically excluded so that days with no precipitation are not included in the analysis of sequences with identical data.

For temperatures, the gap filling and homogenization were carried out following the methodology of Varotsos et al. (2023b). This method reconstructs missing daily temperatures values (*TX*, *TG* and *TN*) over an extended period of time, using climatol R package, station data and the ERA5-Land reanalysis dataset (Muñoz-Sabater et al., 2021). Since this method is capable of reconstructing temperatures both forward and backward in time, it was selected to provide consistent and homogenized data for the period 1981–2019 across all 122 available stations. For further details on the methodology, the reader is referred to the work of Varotsos et al. (2023b).

For precipitation, gap filling and homogenisation were carried out in two phases. In the first phase, stations covering the period 1981–2019 (including HNMS stations, the Ministry of Environment and Energy network, and the historical Thissio NOA station) were post-processed using climatol package, with data from the nearest station serving as the reference. In the second phase, the homogenised data series for the period 2010–2019 were used to fill gaps and homogenise the daily precipitation data of the NOAAN network. Following these procedures, the total number of precipitation data series is 264 for 1981–2010 and 312 for 2010–2019.

## 2.3 WRF simulations

For the atmospheric simulations, the Advanced Weather Research and Forecasting Model (WRF-ARW) version 4.1.3 (Skamarock et al., 2019) was employed. WRF-ARW is a limited-area atmospheric model based on a fully compressible, non-hydrostatic dynamic core. Vertically, it utilizes terrain-following, mass-based hybrid sigma-pressure coordinates based on dry hydrostatic pressure, with support for vertical grid stretching. Horizontally, the model applies an Arakawa C-grid staggering.

WRF is widely used in both operational forecasting (Sofia et al., 2024; Patlakas et al., 2023) and scientific research (Pantillon et al., 2024; Patlakas et al., 2024; Politi et al., 2021; Stathopoulos et al., 2023; Otero-Casal et al., 2019). These studies include comprehensive evaluations of the model's performance not only over the present study area but also in regions with similar topographic and climatic characteristics, demonstrating its reliability in representing climatological fields. In this analysis, the WRF model was configured with three two-way nested grids to adequately capture both regional and local-scale processes. The coarser one has a resolution of 9 km, covering a large area that includes parts of North Africa and Central Europe. The inner grids are focused on the Eastern Mediterranean and Greece, with spatial resolutions of 3 km and 1 km, respectively (Fig. 2). Vertically, the model consists of 48 layers.

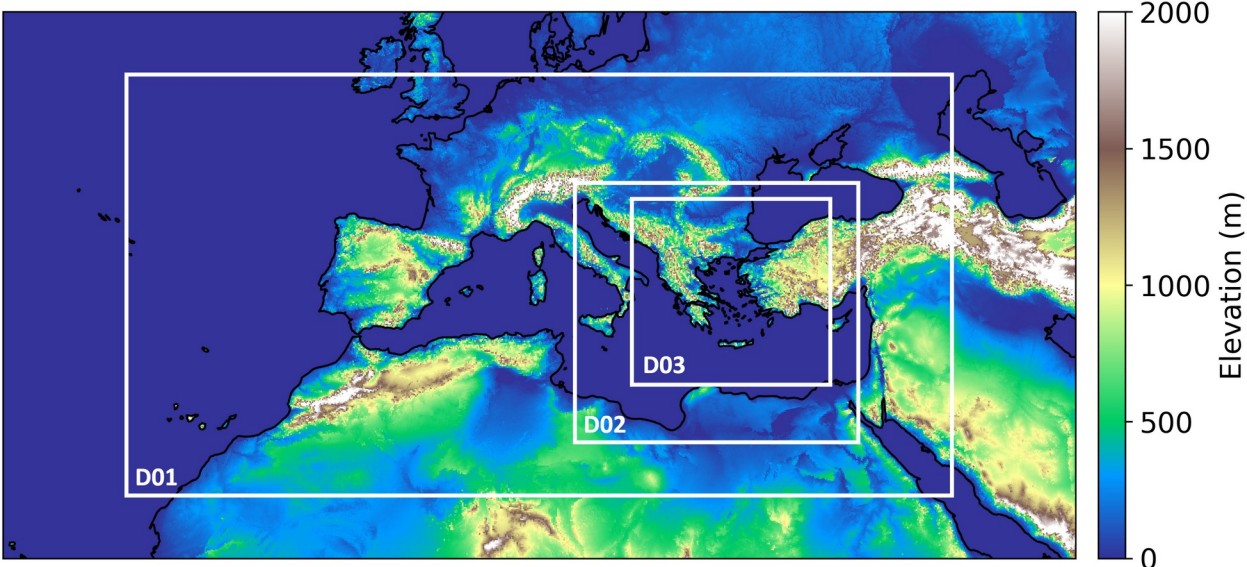

**Figure 2: WRF-ARW model domains.**

The main physics options and parameterizations used are summarized in the next table (Table 1).

**Table 1: WRF model physical schemes and properties.**

| | |
|---|---|
| Microphysics | Thompson scheme (Thompson et al., 2004) |
| Cumulus Parameterization | Kain–Fritsch scheme (Kain, 2004) |
| Long wave radiation physics | RRTMG scheme (Iacono et al., 2008) |
| Short wave radiation physics | RRTMG scheme |
| Planet boundary layer | Yonsei University (YSU) PBL scheme (Hong et al., 2006) |
| Surface layer option | Monin–Obukhov similarity scheme |
| Land-surface physics | Thermal Diffusion scheme |

For the initial and the boundary conditions, the ERA-5 (Hersbach et al., 2020) hourly data has been incorporated. This is the latest global atmospheric reanalysis product produced by the European Centre for Medium-Range Weather Forecasts (ECMWF), covering the period from 1940 to the present with continuous real time updates and a spatial resolution of 0.25 degrees. Terrain elevation data is obtained from the ASTER Global Digital Elevation Map (GDEM) from United States Geological Survey (USGS, Slater et al., 2011) with a resolution of 30 m, and land use information from the Coordination of Information on the Environment (CORINE, CLMS, 2018) at a 250 m resolution.

Following the approach of Varotsos et al. (2023a), the year selected for the WRF simulation was chosen based on its mean monthly annual cycle, the lowest deviations from its long-term mean for the period 1981–2019. The analysis revealed that the year 1999 had the lowest temperature and precipitation deviations, from the long-term mean. It should be noted here that the selection of the year of the WRF simulation is not of primary importance in this study, since it is used as a physically based spatial interpolator, as described in Sect. 3.2. Therefore, the key requirement is that the WRF model provides a continuous and physically consistent representation of the temperature field across the region's complex terrain, a capability supported by the aforementioned studies.

## 3 Methodology

The methodology applied in this study to produce the daily gridded observational precipitation dataset for Greece for the years 1981–2019, aligns to that used in the early versions of E-OBS (Haylock et al., 2008) and IBERIA01 (Herrera et al., 2019). For temperature variables we adopted and extended the methodology described by Varotsos et al. (2023a), where the available observed data were blended with WRF output through gridding techniques. While Varotsos et al. (2023a) focused on Attica region, we expanded their methodology to cover the entire Greek territory.

### 3.1 Spatiotemporal modeling for precipitation

The steps to obtain the daily grids for precipitation are as follows:

- interpolation of monthly totals (12 values x 39 years) using altitude as a covariate to account for altitude dependencies (station altitude for modeling and the Global Multi-resolution Terrain Elevation Data 2010 (GMTED2010), 30 arcsec version, altitude for interpolation).

The following approaches were examined to calculate the monthly precipitation fields:

(i) a "Fixed Rank Kriging approach" (FRK). FRK is a geostatistical interpolation technique that approximates a spatial field using a low-rank representation of the underlying spatial process. It models the spatial covariance structure through a set of basis functions, allowing efficient estimation even with large datasets (Nychka et al., 2015a). In this study, FRK is implemented using the LatticeKrig package (Nychka et al., 2019) in R (R Core Team, 2024), where the model parameters, including variance components and spatial range parameters, are estimated using maximum likelihood estimation.

(ii) Generalized Additive Models (GAM) are a semi-parametric extension of Generalized Linear Models that assume the underlying relationships are additive and smooth. Their primary strength lies in their ability to capture highly non-linear and non-monotonic relationships between the response variable and explanatory variables (Wood, 2017). In this study, monthly precipitation sums are modeled as smooth functions of longitude, latitude and elevation using thin plate regression splines, with smoothing parameters estimated using restricted maximum likelihood (mgcv R package, Wood and Wood, 2015).

(iii) two ML algorithms namely K-Nearest Neighbors (KNN) and Support Vector Machines (based on an exponential radial basis function, SVM). KNN estimates the value of an unknown data point by identifying its k closest neighbors in the spatial dataset, where k is a user-defined hyperparameter (Nwaila et al., 2024). The predicted value is computed as a weighted average of these neighbors' values, with the weights typically based on the distance to the target point, i.e. closer neighbors have greater influence. In this study, k ranges from 2 to 30 in increments of 1. SVM is a ML algorithm, effective in capturing non-linear spatial trends, which seeks a function that predicts the value of an unknown data point, while balancing accuracy and model complexity (Bonsoms and Ninyerola, 2024). The complexity is regulated by the cost parameter C (tested values: 0.1, 1, 5, 10), while the smoothness of the kernel is governed by sigma (tested values: 0.01, 0.025, 0.05, 0.075, 0.1). For both algorithms, optimal hyperparameters (k for KNN, and C and sigma for SVM) are selected using tenfold cross-validation combined with grid search, using the caret package in R (Kuhn, 2008; R Core Team, 2024). The choice of these two ML algorithms was based on literature (e.g., Bonsoms and Ninyerola, 2024), as well as on preliminary tests that included other algorithms, such as random forests, gradient boosting machines and neural networks. However, the latter algorithms were excluded as they produced unrealistic cross-hatched patterns in the precipitation background fields.

- interpolation of the daily anomalies (quotient from the monthly total values, 365 values/year × 39 years) of the observational data. The final step of precipitation interpolation was implemented using an exponential covariance with the covariance parameters estimated through maximum likelihood. For more information, the reader is referred to the fields R package spatialProcess function (Nychka et al., 2015b). To obtain the final daily gridded dataset for precipitation, the two interpolated precipitation products obtained are superimposed by multiplication.

## 3.2 Spatiotemporal modeling for temperatures

As outlined earlier in Sect. 3, the methodology of Varotsos et al. (2023a) was applied to generate the daily gridded temperature datasets with the key steps as implemented in this work briefly summarized below.

The approach is implemented in four steps. The first two steps involve the WRF perturbation to align with observed long-term climate temporal characteristics while preserving its spatial variability. This is achieved by adding the interpolated monthly biases, calculated as the difference between the mean monthly annual cycles over the period 1981–2019 and the monthly means calculated over the year 1999 at the closest WRF grid point to each station location, to the mean monthly values at each grid point. The third step involves the construction of the gridded dataset following the first two steps mentioned for the precipitation dataset by adding the interpolated mean monthly values, derived by the different methods (FRK, GAM, KNN, SVM) with the interpolated daily anomalies calculated as the difference between the daily observation and the monthly mean. The last step involves the transfer of perturbed WRF output to the daily gridded dataset of the previous step using the unbiasing bias adjustment method (Déqué, 2007).

The methodology presented in this study regarding the gridding of temperature data is flexible and allows for the integration of other regional datasets (e.g. the Copernicus regional reanalysis for Europe, CERRA) in multiple ways, depending on the objective. For example, if the aim is to develop a gridded dataset at a resolution similar to that of CERRA (5.5 km × 5.5 km), the WRF output could be replaced entirely with CERRA data. Alternatively, a combined approach could be employed, whereby CERRA is used in conjunction with WRF output to produce a statistically downscaled CERRA dataset, which can then be bias-adjusted using observational data. More specifically, in the first step of the methodology, observational data could be substituted with CERRA values at the nearest grid points to the stations locations. These values would be used to perturb the WRF output, followed by application of the final step of the methodology to a 1 km regridded version of the CERRA dataset. The resulting high-resolution dataset could then be bias-adjusted by adding the interpolated mean monthly differences between the station observations and the corresponding values from the 1 km CERRA dataset.

## 3.3 Evaluation analysis against withheld station data

An evaluation of the different approaches for constructing the daily grids was conducted for the test period (2010–2019). The aim of this evaluation was to identify the most effective method for generating daily datasets for all variables over the period 1981–2019. To achieve this, approximately 10 % of the stations were excluded from the dataset, and the interpolated values were compared with the observed values at those locations. For both temperatures and precipitation station data, the withheld stations were selected using the minimax distance design (Johnson et al., 1990; Cornes et al., 2018), which minimizes the maximum distance from each withheld station to any of the other stations (Fig. 1a and 1b).

The accuracy of the various approaches is assessed using Bias (BIAS), Root Mean Square Error (RMSE), Mean Absolute Error (MAE), and Kling-Gupta Efficiency (KGE) measures. The KGE metric is a measure of goodness of fit that has been routinely used to evaluate climate datasets (Beck et al., 2019; Bhuiyan et al., 2019; Avila-Diaz et al., 2021). KGE can obtain

values ranging from -Inf to +1, with values of +1.0 and −1.0 to indicate perfect positive and negative linear correlation between the reference and the assessed timeseries, respectively, while a value of 0 implies no correlation. In this study, gridding techniques are more accurate when KGE values are close to one. When calculating the KGE, three key factors are considered: (i) the Pearson product-moment correlation coefficient (R), (ii) the ratio of the mean of the reconstructed values to the mean of the observed values (beta), and (iii) the variability ratio based on the standard deviations of the reconstructed values to the observed values (alpha).

## 3.4 Comparison against CHELSA-W5E5

The final produced daily gridded datasets for temperatures and precipitation were compared against the corresponding variables from CHELSA-W5E5 v1 (hereafter CHELSA, Karger et al., 2023). CHELSA is a global land dataset providing daily air temperature, precipitation, and downwelling shortwave solar radiation at a 1 km resolution for the period 1979–2016. It is produced by spatially downscaling the 0.5° W5E5 dataset onto a grid based on the GMTED2010 Digital Elevation Model, which is also used in this study. Notably, both the CLIMADAT-GRid and CHELSA are constructed using the same digital elevation model thus sharing the same grid while the shared elevation model ensures consistency in elevation values across corresponding grid points in the two datasets. Moreover, Papa and Koutroulis (2025) found that CHELSA is one of the two more reliable gridded datasets for describing the precipitation dynamics in Greece. The comparison was performed by examining average annual and seasonal means for all temperature variables (sums for precipitation), as well as selected indices from the Expert Team on Climate Change Detection and Indices (ETCCDI, Zhang et al., 2011). These are the number of days with daily $TX > 25$ °C (SU) and $TX > 35$ °C (SU35) for $TX$, the number of days with daily $TN > 20$ °C (TR) for $TN$ and the number of days with $PR > 1$ mm (RR1) as well as the number of days with $PR \geq 10$ mm (RR10) for $PR$. These indices were selected considering the primary climatic characteristics of the studied area, which exhibits Mediterranean-type climate conditions with moderate winters and warm to hot summers.

## 4 Results

### 4.1 Evaluation against withheld station data for the test period (2010–2019)

### 4.1.1 Precipitation results

The values of BIAS, RMSE, MAE and KGE are presented in Table 2 for the precipitation grids produced using the different approaches described in Sect. 3.1. These results are evaluated against withheld observations on both an annual and seasonal scale over the test period. From Table 2, it is evident that on the annual scale, all four approaches exhibit strong and relatively consistent performance across all evaluation metrics. BIAS values are minimal, indicating that none of the models significantly overestimate or underestimate precipitation. FRK shows the smallest annual BIAS (−0.01 mm), closely followed by GAM (−0.02 mm) and SVM (−0.05 mm), while KNN shows a negative BIAS of −0.08 mm. Regarding RMSE

and MAE, it is shown that the different approaches yield similar values across both statistical measures with annual RMSE (MAE mm) being lower than 1.45 (0.65 mm), while KGE values are higher than 0.85 for all approaches, with SVM and FRK reaching the highest values 0.93 and 0.92, respectively. At the seasonal scale, greater variability is shown. During winter (DJF, December-January-February), SVM and GAM tend to significantly underestimate precipitation, with SVM showing the most negative BIAS (−0.16 mm) and a KGE of 0.91, while GAM also underperforms with a lower KGE of 0.81 and relatively high MAE (1.01 mm). In contrast, KNN tends to overestimate during this period (BIAS = 0.12 mm), but still maintains a moderate KGE of 0.84. FRK, once again, shows stability with low BIAS (−0.02 mm), competitive RMSE (1.51 mm), and a strong KGE of 0.88. During spring (MAM, March-April-May) and summer (JJA, June-July-August), all approaches show better agreement with observations. FRK continues to perform robustly with the lowest RMSE (MAE) values (0.94 mm (0.50 mm) for MAM and 0.65 mm (0.29 mm) for JJA), and high KGE values (0.90 and 0.82, respectively). The rest of the approaches perform reasonably, though GAM's MAE remains higher in MAM, and SVM and KNN show slight deviations in JJA. In autumn (SON, September-October-November), all approaches tend to overestimate precipitation, with KNN showing the highest BIAS (0.16 mm). SVM shows the lowest BIAS (0.02 mm) and maintains a KGE of 0.94, the highest among the methods for this season. Nevertheless, FRK demonstrates consistent performance with a balanced BIAS (0.05 mm), relatively low RMSE and MAE of 1.27 mm and 0.66 mm, respectively, and a strong KGE of 0.93. Overall, the results indicate FRK as the most stable and reliable method, delivering consistently low BIAS and error across both annual and seasonal scales while maintaining high KGE values throughout the year.

In Fig. 3 the average annual daily distributions for precipitation are shown for the 10 years test period. The results indicate that the methods can capture the annual total precipitation, with the maximum relative absolute biases of 8 % or less across all methods.

Table 2: Precipitation annual and seasonal BIAS, RMSE, MAE and KGE statistics based on daily values between the 30 reference station and the interpolated ones as derived from the different interpolation approaches.

| | FRK | | | | GAM | | | | KNN | | | | SVM | | | |
|---|---|---|---|---|---|---|---|---|---|---|---|---|---|---|---|---|
| *PR* | BIAS | RMSE | MAE | KGE | BIAS | RMSE | MAE | KGE | BIAS | RMSE | MAE | KGE | BIAS | RMSE | MAE | KGE |
| Annual | -0.01 | 1.13 | 0.60 | 0.92 | -0.02 | 1.45 | 0.63 | 0.87 | -0.08 | 1.18 | 0.62 | 0.87 | -0.05 | 1.09 | 0.59 | 0.93 |
| DJF | -0.02 | 1.51 | 0.95 | 0.88 | -0.06 | 1.46 | 1.01 | 0.81 | 0.12 | 1.56 | 0.97 | 0.84 | -0.16 | 1.43 | 0.91 | 0.91 |
| MAM | 0.03 | 0.94 | 0.50 | 0.90 | -0.03 | 1.47 | 0.93 | 0.88 | -0.11 | 0.99 | 0.54 | 0.86 | -0.04 | 0.92 | 0.51 | 0.92 |
| JJA | -0.09 | 0.65 | 0.29 | 0.82 | -0.16 | 1.47 | 0.3 | 0.70 | -0.06 | 0.63 | 0.29 | 0.86 | -0.08 | 0.67 | 0.31 | 0.82 |
| SON | 0.05 | 1.27 | 0.66 | 0.93 | 0.07 | 1.43 | 0.71 | 0.87 | 0.16 | 1.34 | 0.69 | 0.85 | 0.02 | 1.22 | 0.66 | 0.94 |

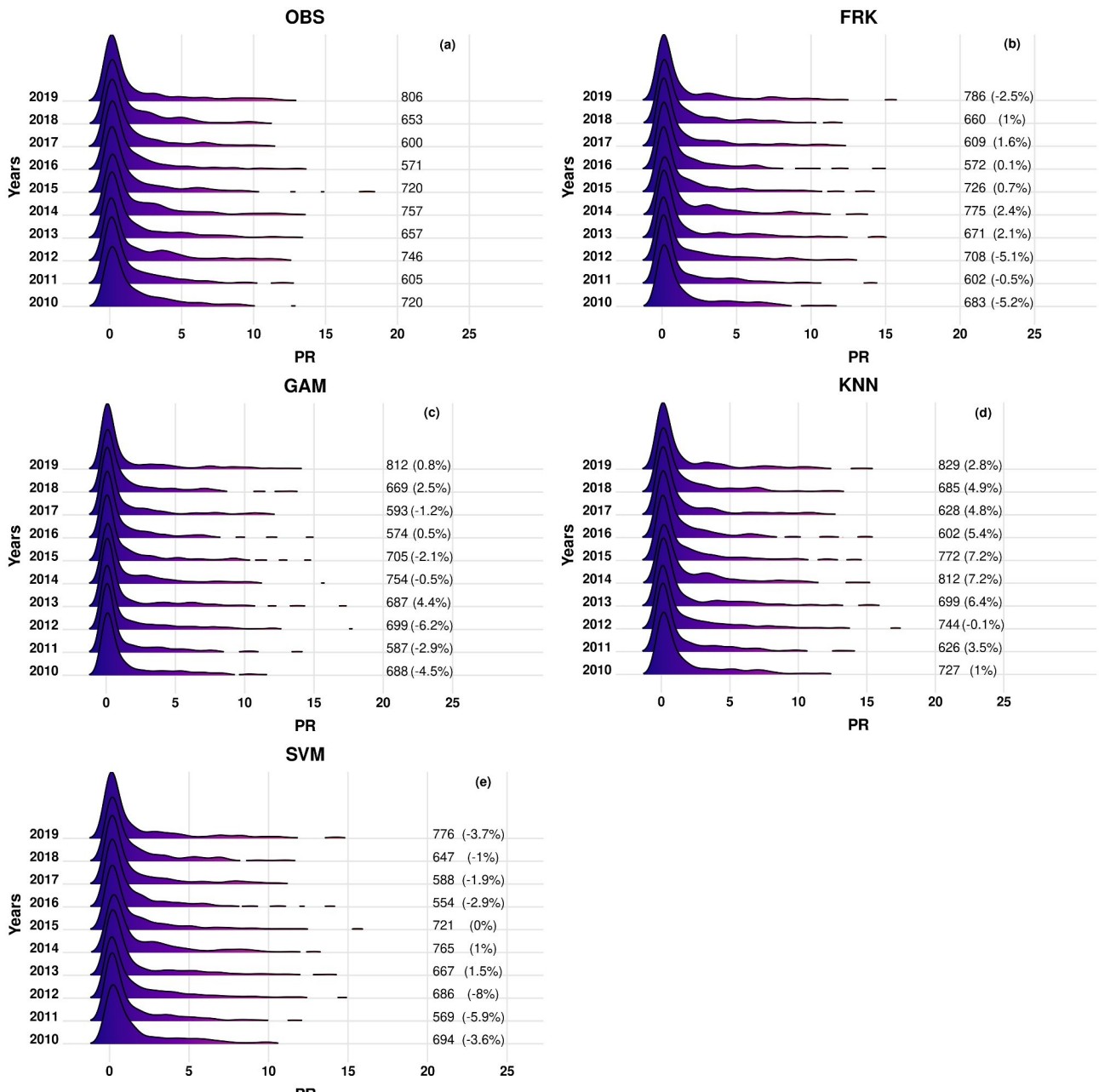

**Figure 3. Density distributions of daily precipitation values over the withheld station data for the observations (OBS) and the different methods used for interpolation for the years 2010–2019. The values shown in the plots are the total annual precipitation values whereas the relative biases between the different methods and the observations are shown in parenthesis.**

When all available stations are used for the interpolation for the period 2010–2019 (Fig. 4), the results indicate that the west to east gradient of precipitation in Greece, which exhibits the largest and lowest amounts of yearly precipitation (Gofa et al.,

2019) is captured by all approaches. However, as it is evident from Fig. 4, FRK maintains strong predictive skill when applied to the full set of available stations, with only a modest increase in error, reflecting its ability to handle spatial heterogeneity and non-stationarity common in precipitation fields. GAM error metrics increase somewhat on the full dataset, however, retaining a reasonable predictive skill, indicating its capacity to capture important spatial patterns. For KNN, an opposite sign BIAS is evident when compared to the results of Table 2 indicating that the method is strongly dependent on

the proximity to known data. As a result, the method exhibits poorer extrapolation ability than FRK and GAM. In contrast, SVM shows a significant decline in performance when applied to the full station network, as it lacks explicit modeling of spatial structures despite its capability to capture complex nonlinear relationships (Heinke et al. 2023), resulting in higher errors and biases across diverse climatic and geographic regions. This behavior is highlighted in the mountainous area in Crete where total annual precipitation is much lower than the other methods.

Overall, this comparison highlights that FRK is better suited for robust precipitation interpolation across regions with complex topography.

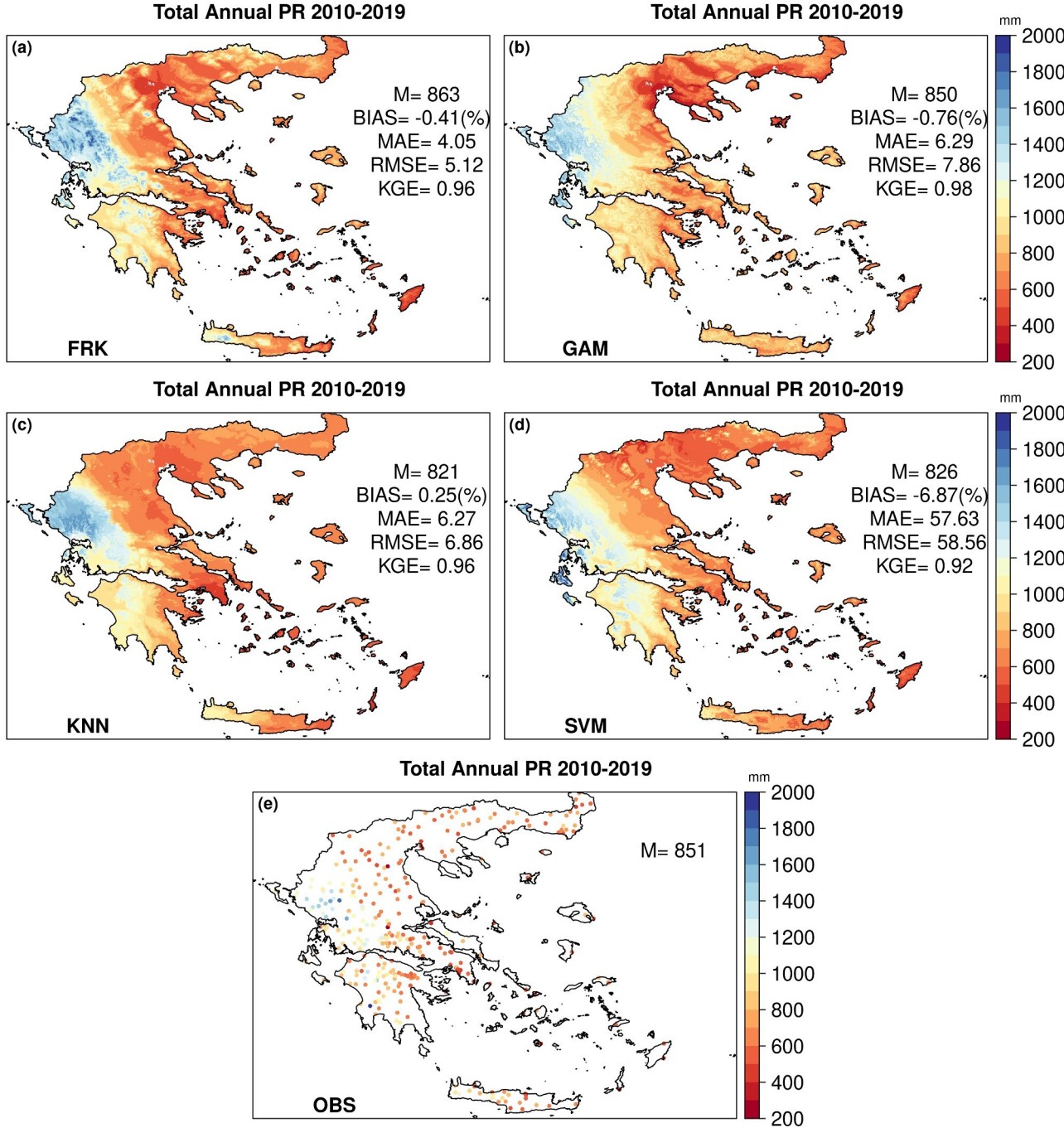

**Figure 4. Spatial distribution of total annual precipitation for the period 2010–2019, as estimated by the different interpolation methods (panels a–d) and observed data (panel e). Each panel includes the spatial average (M) calculated over all grid points (or over all stations in panel e). For panels a–d, the relative BIAS, MAE (in mm), RMSE (in mm), and KGE are provided, based on comparisons between the interpolated values at the nearest grid points and the corresponding station observations.**

## 4.1.2 Temperature results

This section is dedicated to the evaluation of the effectiveness of the proposed approaches in reproducing observed temperatures, specifically addressing the third phase of the temperature methodology described in Sect. 3.2.

From Table 3 and Fig. 5-7, it is evident that all methods perform well in capturing the temporal temperature characteristics for $TX$, $TG$ and $TN$. Table 3 presents the values of the metrics as calculated from daily values, which offer insight into the different methods systematic errors at the finer temporal scale. For $TX$, KNN and SVM exhibit the best overall performance across RMSE, MAE, and KGE, with seasonal RMSE values consistently below 0.67 °C and high KGE values ($\geq 0.93$). KNN performs particularly well in colder seasons (DJF and SON), while SVM shows better results in warmer periods (MAM and JJA). FRK ranks third, showing competitive RMSE and MAE values, though it consistently underestimates $TX$ with a negative bias across all seasons, most notably in SON (−0.47 °C). GAM, while displaying the highest RMSE and MAE values in every season, has the lowest annual BIAS (0.00 °C) and relatively low seasonal BIAS values (e.g., −0.04 °C in DJF, −0.27 °C in MAM), indicating alignment with the observed annual mean but poor performance in capturing daily variability. When examining the annual differences between the observations and the various methods (Fig. 5), GAM demonstrates the lowest absolute annual deviations, remaining below 0.2 °C, indicating strong agreement with long-term averages. In contrast, FRK, exhibits the highest annual deviations, reaching up to 0.6 °C.

For $TG$ and $TN$, FRK emerges as the most robust method, outperforming others across all metrics on both annual and seasonal scales. It consistently delivers the lowest RMSE, MAE, and KGE values. Importantly, FRK also exhibits the lowest annual BIAS values for $TG$ (0.08 °C) and $TN$ (0.10 °C), underscoring its minimal systematic deviation from observations. In comparison, other methods present substantially higher BIAS values, particularly KNN (0.57 °C for $TG$; 0.74 °C for $TN$) and SVM (0.71 °C for $TG$; 0.56 °C for $TN$). Furthermore, FRK's mean absolute annual deviations remain lower than 0.3 °C for both $TG$ and $TN$ (Fig. 6-7), whereas other methods show deviations reaching up to 0.8 °C, depending on the variable and method.

**Table 3: Daily maximum (*TX*), mean (*TG*) and minimum (*TN*) temperatures annual and seasonal BIAS, RMSE, MAE and KGE statistics based on daily values between the 13 reference station and the interpolated ones as derived from the different interpolation approaches.**

| | FRK | | | | GAM | | | | KNN | | | | SVM | | | |
|---|---|---|---|---|---|---|---|---|---|---|---|---|---|---|---|---|
| | BIAS | RMSE | MAE | KGE | BIAS | RMSE | MAE | KGE | BIAS | RMSE | MAE | KGE | BIAS | RMSE | MAE | KGE |
| *TX* | | | | | | | | | | | | | | | | |
| Annual | -0.33 | 0.69 | 0.55 | 0.98 | 0.00 | 1.63 | 0.48 | 0.98 | 0.11 | 0.60 | 0.50 | 0.98 | -0.12 | 0.60 | 0.49 | 0.98 |
| DJF | -0.41 | 0.68 | 0.55 | 0.91 | -0.04 | 1.6 | 0.41 | 0.93 | -0.30 | 0.60 | 0.49 | 0.93 | -0.38 | 0.64 | 0.53 | 0.94 |

| | | | | | | | | | | | | | | | | |
|---|---|---|---|---|---|---|---|---|---|---|---|---|---|---|---|---|
| MAM | -0.06 | 0.60 | 0.47 | 0.97 | -0.27 | 1.53 | 0.53 | 0.97 | -0.33 | 0.67 | 0.56 | 0.98 | -0.21 | 0.62 | 0.51 | 0.98 |
| JJA | -0.40 | 0.72 | 0.58 | 0.97 | -0.14 | 1.35 | 0.49 | 0.98 | -0.18 | 0.60 | 0.50 | 0.98 | -0.12 | 0.56 | 0.46 | 0.98 |
| SON | -0.47 | 0.76 | 0.61 | 0.96 | -0.15 | 1.48 | 0.47 | 0.98 | -0.02 | 0.55 | 0.45 | 0.99 | 0.03 | 0.57 | 0.46 | 0.98 |
| *TG* | | | | | | | | | | | | | | | | |
| Annual | 0.08 | 0.45 | 0.36 | 0.98 | 0.41 | 2.81 | 0.48 | 0.97 | 0.57 | 0.69 | 0.59 | 0.97 | 0.71 | 0.82 | 0.73 | 0.96 |
| DJF | -0.18 | 0.45 | 0.36 | 0.98 | 0.24 | 2.56 | 0.38 | 0.97 | 0.53 | 0.67 | 0.56 | 0.93 | 0.64 | 0.77 | 0.66 | 0.92 |
| MAM | 0.23 | 0.45 | 0.36 | 0.97 | 0.53 | 2.43 | 0.57 | 0.96 | 0.66 | 0.75 | 0.67 | 0.95 | 0.70 | 0.80 | 0.71 | 0.95 |
| JJA | 0.23 | 0.48 | 0.37 | 0.97 | 0.51 | 2.32 | 0.55 | 0.97 | 0.53 | 0.66 | 0.57 | 0.97 | 0.78 | 0.88 | 0.79 | 0.96 |
| SON | 0.23 | 0.42 | 0.35 | 0.96 | 0.51 | 2.53 | 0.43 | 0.97 | 0.53 | 0.67 | 0.58 | 0.97 | 0.78 | 0.83 | 0.74 | 0.96 |
| *TN* | | | | | | | | | | | | | | | | |
| Annual | 0.10 | 0.52 | 0.41 | 0.99 | 0.51 | 3.85 | 0.59 | 0.96 | 0.74 | 0.88 | 0.77 | 0.94 | 0.56 | 0.77 | 0.64 | 0.95 |
| DJF | -0.04 | 0.63 | 0.52 | 0.92 | 0.39 | 3.85 | 0.58 | 0.88 | 0.75 | 0.97 | 0.80 | 0.80 | 0.66 | 0.93 | 0.75 | 0.82 |
| MAM | 0.22 | 0.52 | 0.43 | 0.98 | 0.66 | 3.85 | 0.70 | 0.93 | 0.75 | 0.88 | 0.78 | 0.92 | 0.49 | 0.72 | 0.60 | 0.95 |
| JJA | 0.17 | 0.42 | 0.34 | 0.98 | 0.55 | 3.79 | 0.57 | 0.97 | 0.73 | 0.83 | 0.73 | 0.96 | 0.49 | 0.65 | 0.54 | 0.95 |
| SON | 0.06 | 0.48 | 0.38 | 0.99 | 0.45 | 3.85 | 0.54 | 0.96 | 0.74 | 0.88 | 0.77 | 0.94 | 0.62 | 0.79 | 0.68 | 0.95 |


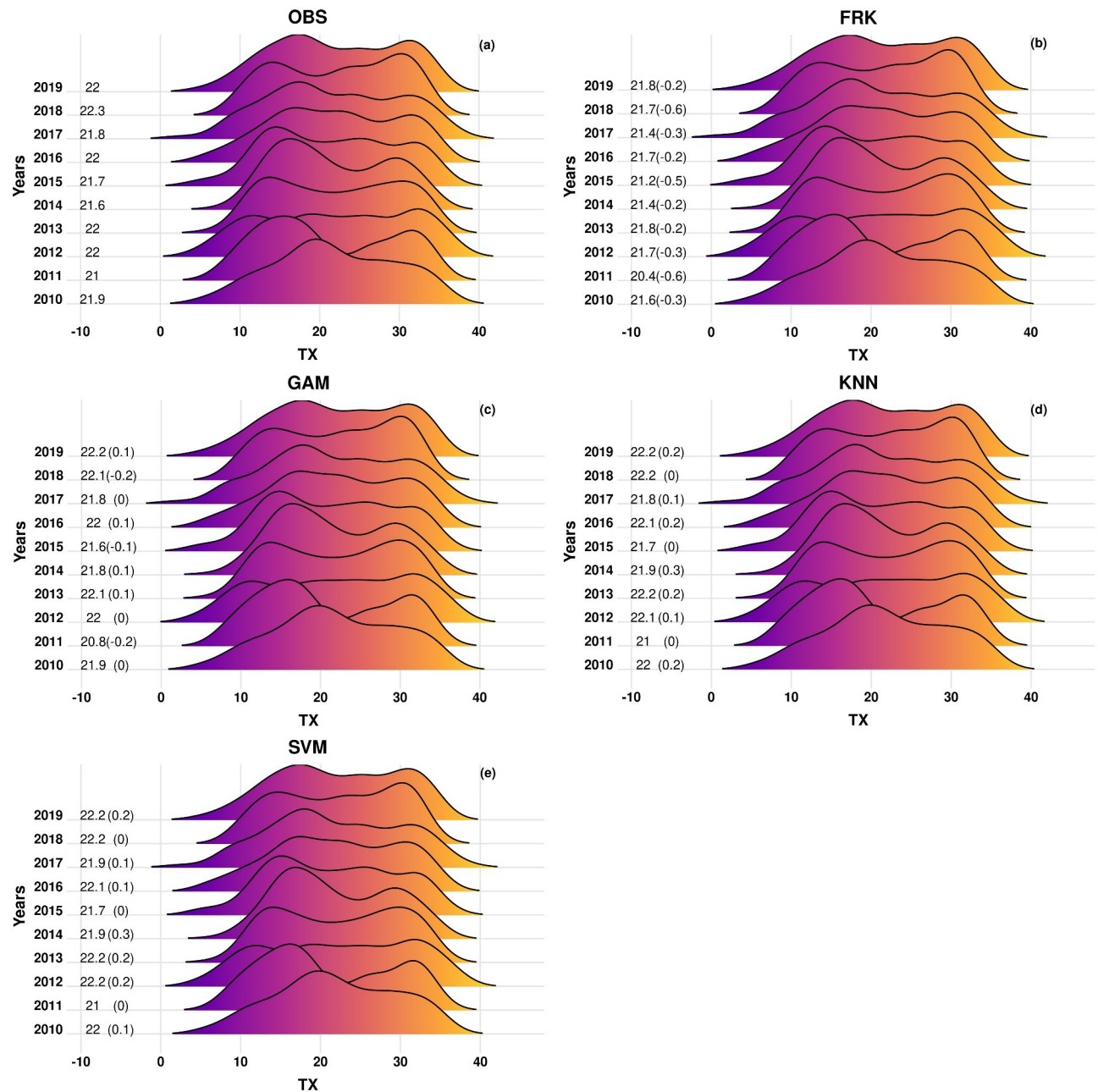

**Figure 5. Density distributions of daily maximum temperature (*TX*) values over the withheld station data for the observations (OBS) and the different methods used for interpolation for the years 2010–2019 The values shown in the plots are the average annual values whereas the biases between the different methods and the observations are shown in parenthesis.**


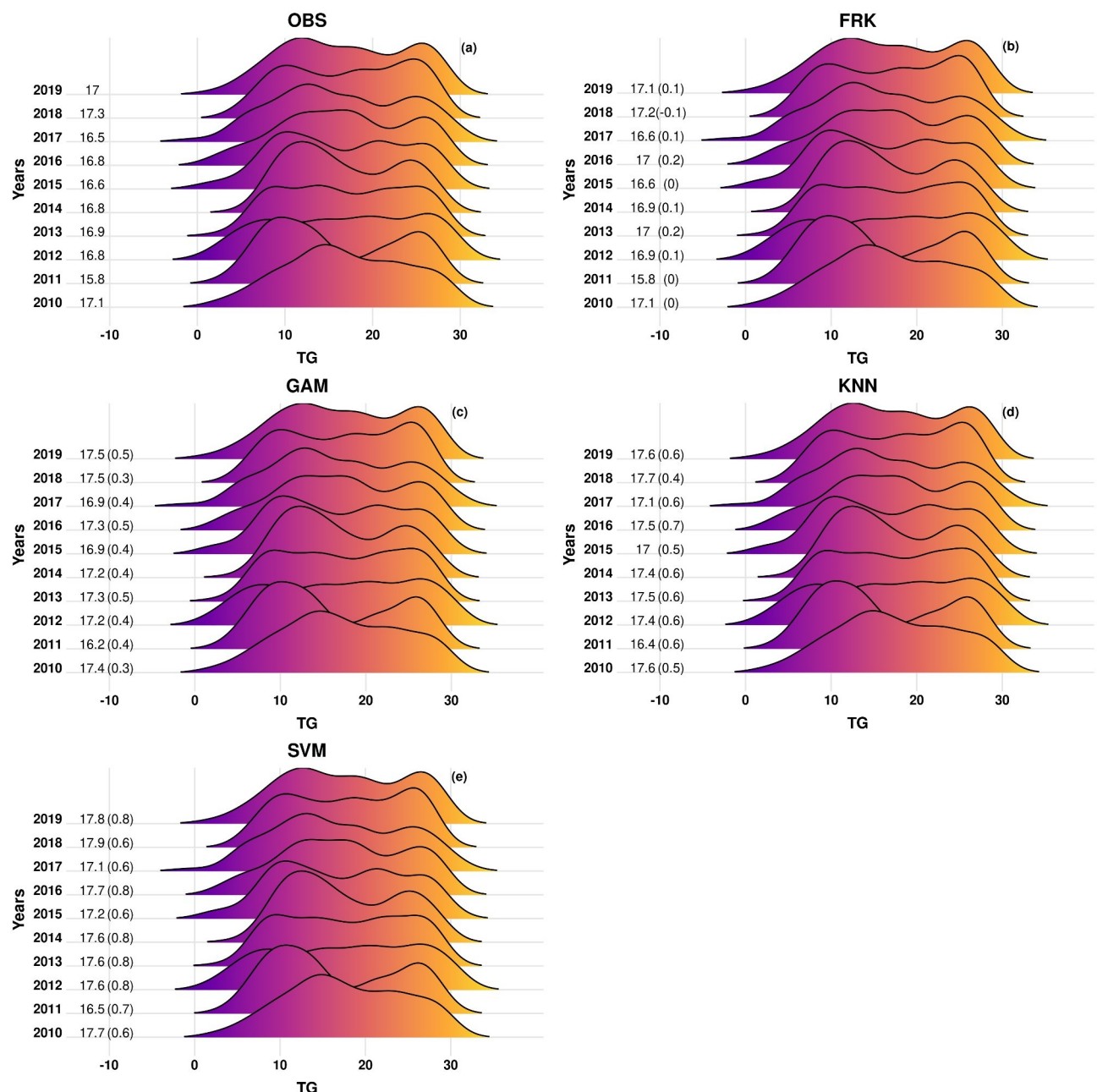

**Figure 6. Density distributions of daily mean temperature (*TG*) over the withheld station data for the observations (OBS) and the different methods used for interpolation for the years 2010–2019 The values shown in the plots are the average annual values whereas the biases between the different methods and the observations are shown in parenthesis.**


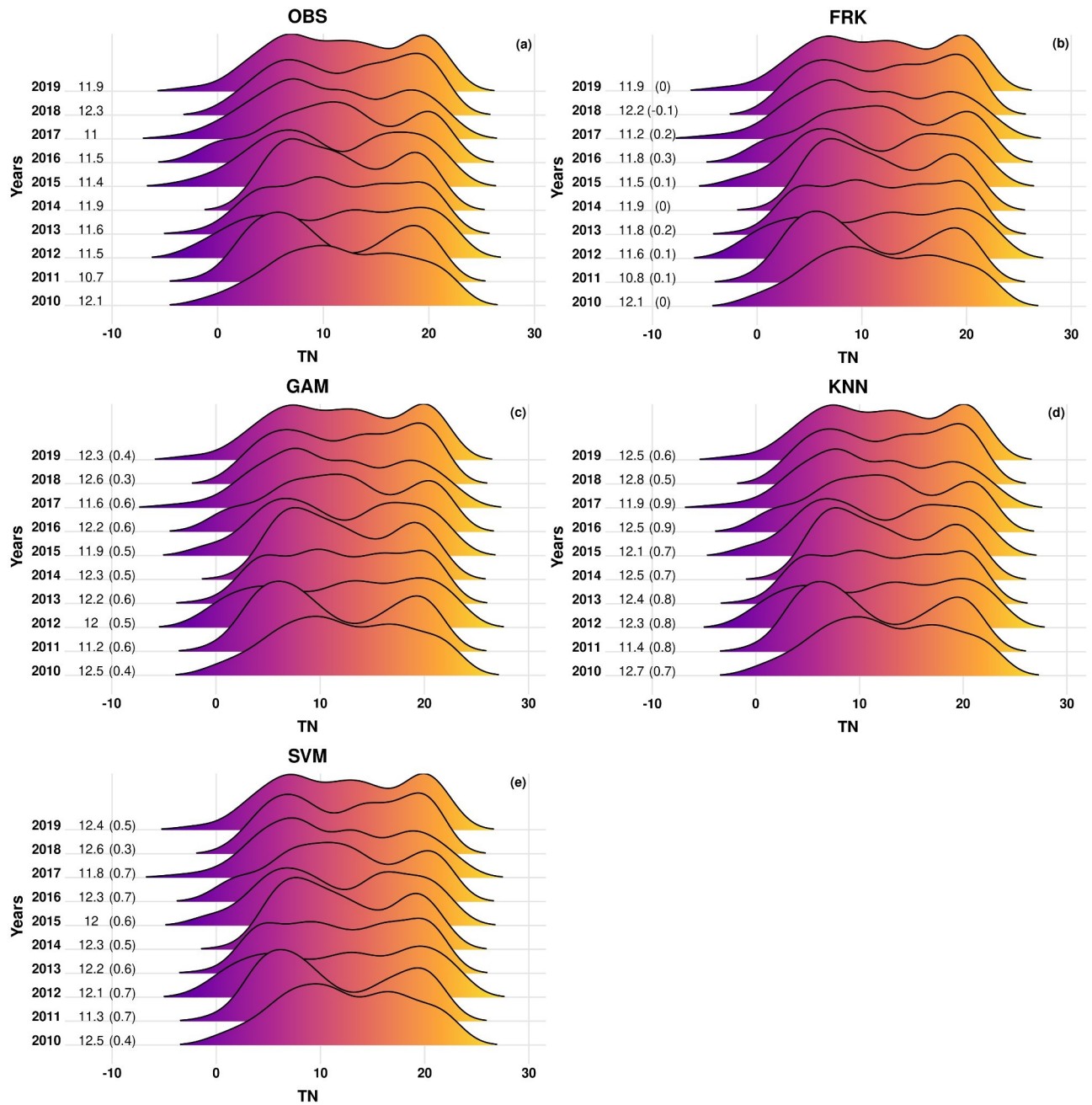

**Figure 7. Density distributions of daily minimum temperature (*TN*) values over the withheld station data for the observations (OBS) and the different methods used for interpolation for the years 2010–2019. The values shown in the plots are the average annual values whereas the biases between the different methods and the observations are shown in parenthesis.**

In terms of spatial distribution, similar patterns to those observed for precipitation are identified. In particular, GAM, KNN
and SVM demonstrate limited ability to accurately represent temperature gradients influenced by topography (not shown).

## 4.2 Results of the comparison between CLIMADAT-GRid against CHELSA for the period 1981–2016

This section presents the results of the comparison between CLIMADAT-GRid and CHELSA for both temperatures and precipitation. It is important to note that, based on the findings in Sect. 4.1, FRK was selected as the method used to construct the CLIMADAT-GRid for both variables.

### 4.2.1 Daily maximum, mean and minimum temperature results

Figures 8, 9 and 10 present the annual and seasonal average temperature results for *TX*, *TG*, and *TN* for the CLIMADAT-GRid and CHELSA datasets, respectively, while their differences are shown in Fig. S1 of the Supplementary Material. For *TX*, both datasets show broadly comparable spatial average temperatures over the entire domain (denoted as M in the figures) (Fig. 8). CLIMADAT-GRid consistently matches station observations well, as indicated by zero BIAS and minimal
error metrics across all seasons. MAE and RMSE values remain at or below 0.02 °C, and KGE values are close to 0.99 throughout. In contrast, CHELSA systematically underestimates *TX*, with biases ranging from −0.49 °C in MAM to −0.69 °C in DJF, and RMSE values up to 0.72 °C. CHELSA's lower KGE values (e.g., 0.86 in DJF) further suggest reduced agreement with observations. CLIMADAT-GRid, blended with WRF in the gridded dataset, captures the orographic temperature gradients more effectively exhibiting lower temperatures in elevated regions such as the northwest of Greece,
the central Peloponnese, and western Crete (Fig. S1). Conversely, CHELSA tends to show cooler conditions in the Ionian and Cycladic islands but warmer conditions in Rhodes and Samos.

For *TG*, the pattern is similar but slightly more pronounced in favor of CLIMADAT-Grid (Fig. 9). The mean annual *TG* is 14.3 °C in CLIMADAT-GRid and 14 °C in CHELSA. CLIMADAT-GRid demonstrates extremely low errors and near-zero bias across all seasons, with RMSE values consistently at 0.01–0.02 °C and KGE values at or near 0.99. In contrast,
CHELSA underestimates *TG* with average annual and seasonal biases between −0.29 °C (DJF) and −0.79 °C (JJA), and RMSE values reaching up to 0.81 °C. The KGE for CHELSA is lower, particularly in DJF (0.90) and SON (0.92), indicating less accurate temperature modeling compared to CLIMADAT-GRid. The spatial differences also reflect the better performance of CLIMADAT-GRid, especially in mountainous regions where it more accurately captures lower mean temperatures (Fig. S1).

For *TN*, both datasets report nearly identical domain-averaged values, with the largest difference being only 0.2 °C in MAM and SON (Fig. 10). CLIMADAT-GRid slightly underestimates *TN* (bias from −0.01 °C to −0.03 °C), whereas CHELSA slightly overestimates it during DJF and MAM (bias up to 0.27 °C), though both datasets perform well in JJA and SON. Despite the small average differences, error metrics again favor CLIMADAT-GRid, which shows low MAE and RMSE (0.01–0.03 °C) and perfect or near-perfect KGE values (≥ 0.99). CHELSA, by contrast, displays larger errors (RMSE up to
0.34 °C in MAM) and lower KGE, particularly annually (0.85) and in JJA (0.85), indicating a modest degradation in its

representation of minimum temperatures. Regionally, the most significant *TN* discrepancies appear in coastal and island areas (Fig. S1). CHELSA shows notably higher *TN* values than CLIMADAT-GRid across Zakynthos, Kefallonia, Crete, and many of the Aegean islands. The higher temperatures in CHELSA can be attributed to the implementation of a basic statistical downscaling approach that employs atmospheric temperature lapse rates, B-spline interpolations, and high-

resolution orography rather than a full physical scheme (Karger et al., 2023). Furthermore, according to the authors, constant lapse rates were utilized for all air temperature variables impacting minimum temperatures to a greater extent, as minimum temperatures in high altitudes are frequently the result of nighttime inversions.

Figure 11 presents a comparative analysis of the number of days exceeding key temperature thresholds, *TX* > 25°C (SU), *TX* > 35°C (SU35), and *TN* > 20°C (TR) based on the CLIMADAT-GRid and CHELSA datasets. While both datasets show

close agreement in domain-averaged values for SU and SU35, notable discrepancies emerge for TR, where CHELSA reports a higher frequency (23 days/year) compared to CLIMADAT-Grid (18 days/year).

When benchmarked against observations, CLIMADAT-GRid demonstrates stronger agreement, particularly for SU. CHELSA underestimates SU by approximately 10 days/year, whereas CLIMADAT-Grid closely tracks observed values, reflecting its higher reliability for this metric. Statistical evaluation supports this, since for SU, CLIMADAT-Grid achieves a

low BIAS (1 °C), MAE (1.11 °C), RMSE (1.4 °C), and a high KGE (0.97), outperforming CHELSA, which shows a significant negative BIAS (−9.34 °C), higher MAE (9.34 °C), RMSE (9.69 °C), and a slightly lower KGE (0.91).

For TR, the results are more nuanced. CLIMADAT-GRid shows a greater negative BIAS (−4.02 days/year), but CHELSA, despite the smaller BIAS (−1.28 days/year), presents higher MAE and RMSE values (3.09 days/year and 3.72 days/years, respectively), suggesting differences in how each dataset captures nighttime temperature extremes. Both datasets perform

comparably in terms of KGE, with values of 0.84 (CLIMADAT-GRid) and 0.82 (CHELSA).

Spatial differences further reveal key distinctions between the two datasets (Fig. S2). Discrepancies in SU are concentrated over the Ionian Sea islands (Zakynthos and Kefallonia) and the Cyclades, while differences in TR are more widespread, notably affecting Crete, the Aegean islands, and again the Ionian Sea region. In addition, CLIMADAT-GRid captures the spatial distribution of TR capturing the urban heat island effect in Athens. In contrast, this urban signature is less pronounced

in the CHELSA data, suggesting limitations in its resolution or calibration over complex urban terrains. Similarly, CLIMADAT-Grid effectively captures the spatial distribution of SU35, accurately identifying thermal hotspots across Greece.

Together, these findings highlight the greater consistency and spatial sensitivity of the CLIMADAT-GRid dataset, particularly in reflecting observed heat extremes and local variability across the Greek region.


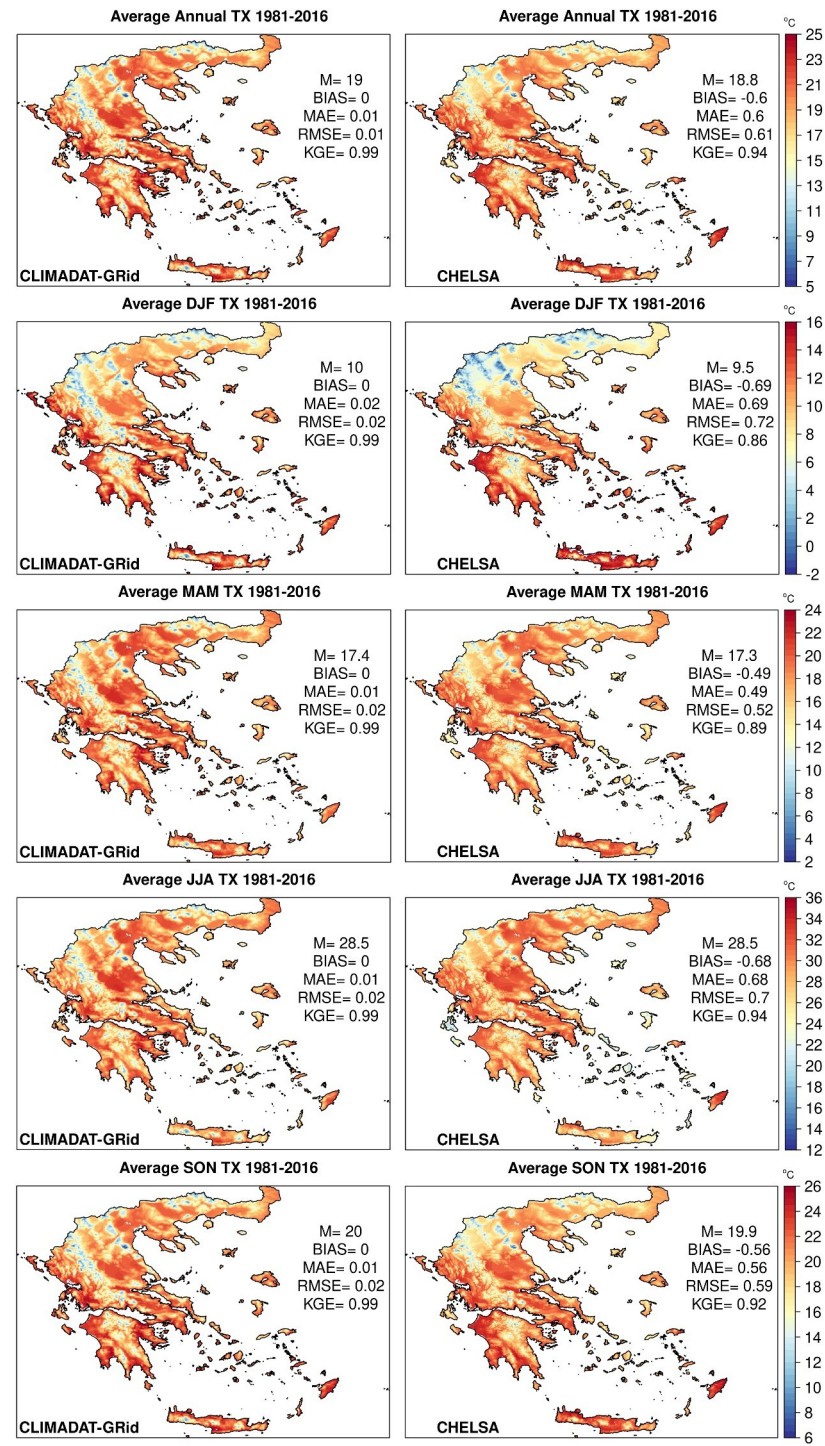

**Figure 8. Average annual and seasonal *TX* (winter (DJF), spring (MAM), summer (JJA) and autumn (SON)) for the period 1981–**
**2016 for CLIMADAT-GRid (left column) and CHELSA (right column). In each panel, M denotes the spatial average over the grid**

points covering the area. In addition, the evaluation metrics between the stations and the data for the closest grid points to the stations locations are shown within each panel.

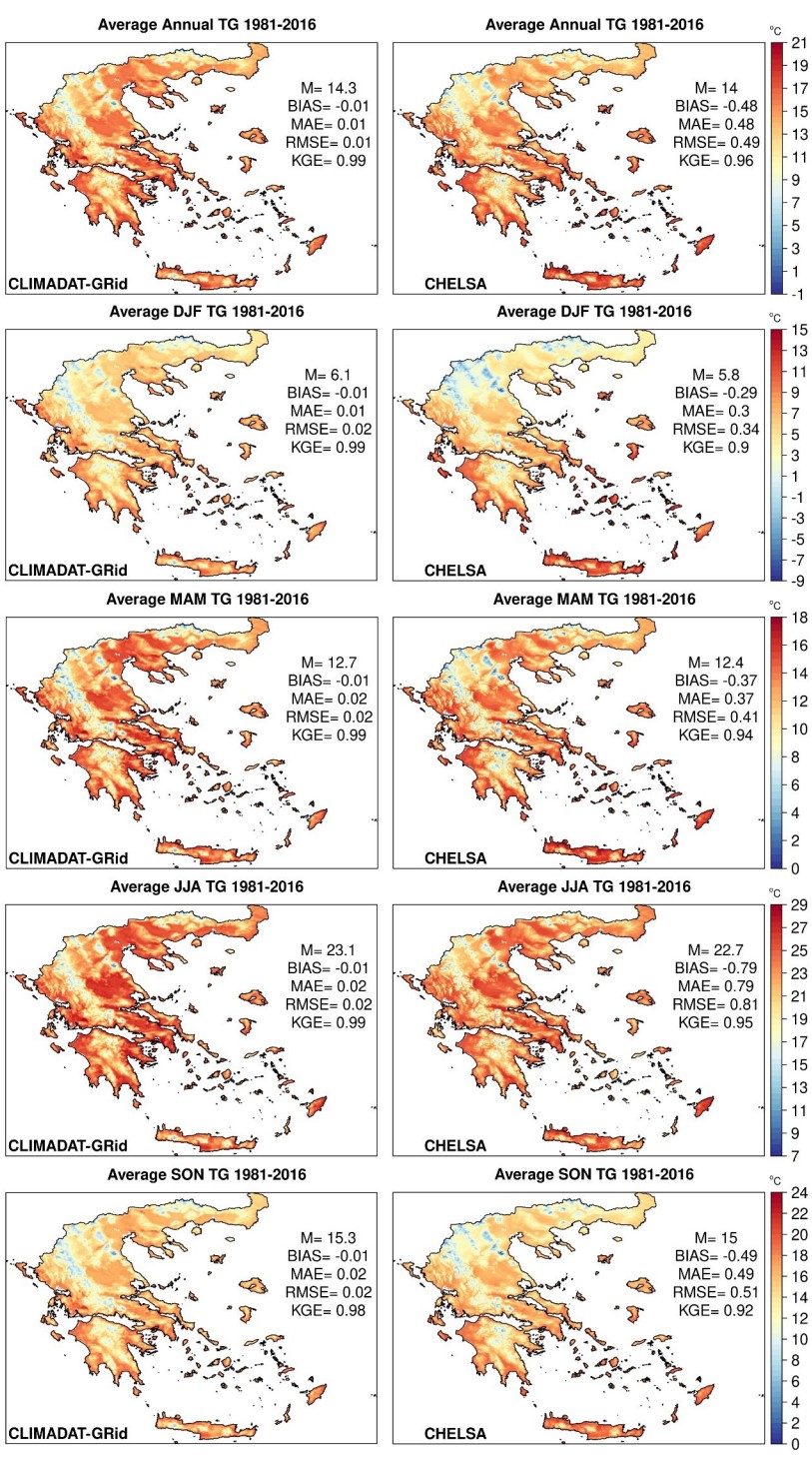

**Figure 9. Average annual and seasonal *TG* (winter (DJF), spring (MAM), summer (JJA) and autumn (SON)) for the period 1981–2016 for CLIMADAT-GRid (left column) and CHELSA (right column). In each panel, M denotes the spatial average over the grid points covering the area. In addition, the evaluation metrics between the stations and the data for the closest grid points to the stations locations are shown within each panel.**

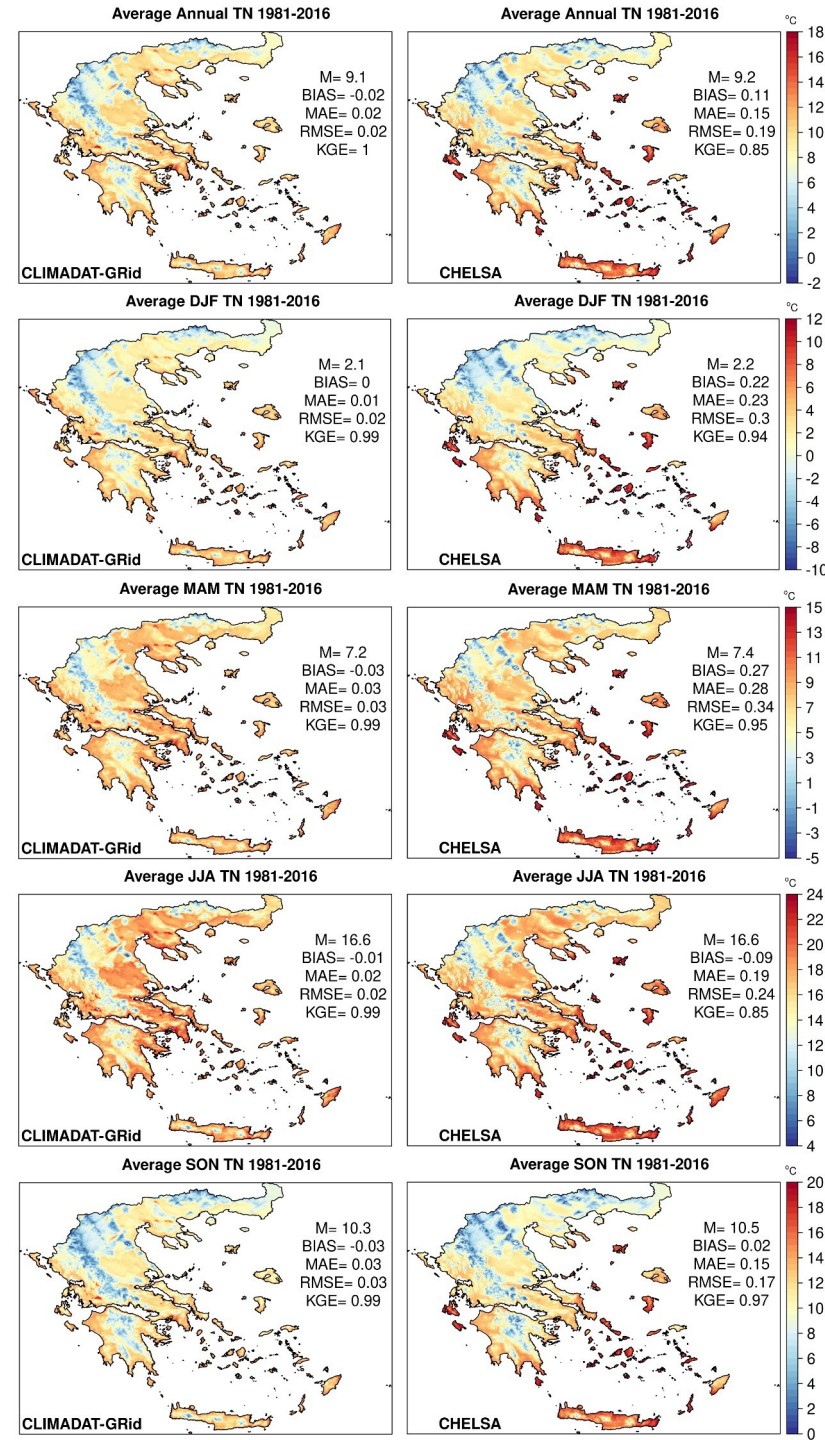

**Figure 10. Average annual and seasonal *TN* (winter (DJF), spring (MAM), summer (JJA) and autumn (SON)) for the period 1981–**
**2016 for CLIMADAT-GRid (left column) and CHELSA (right column). In each panel, M denotes the spatial average over the grid**

points covering the area. In addition, the evaluation metrics between the stations and the data for the closest grid points to the stations locations are shown within each panel.

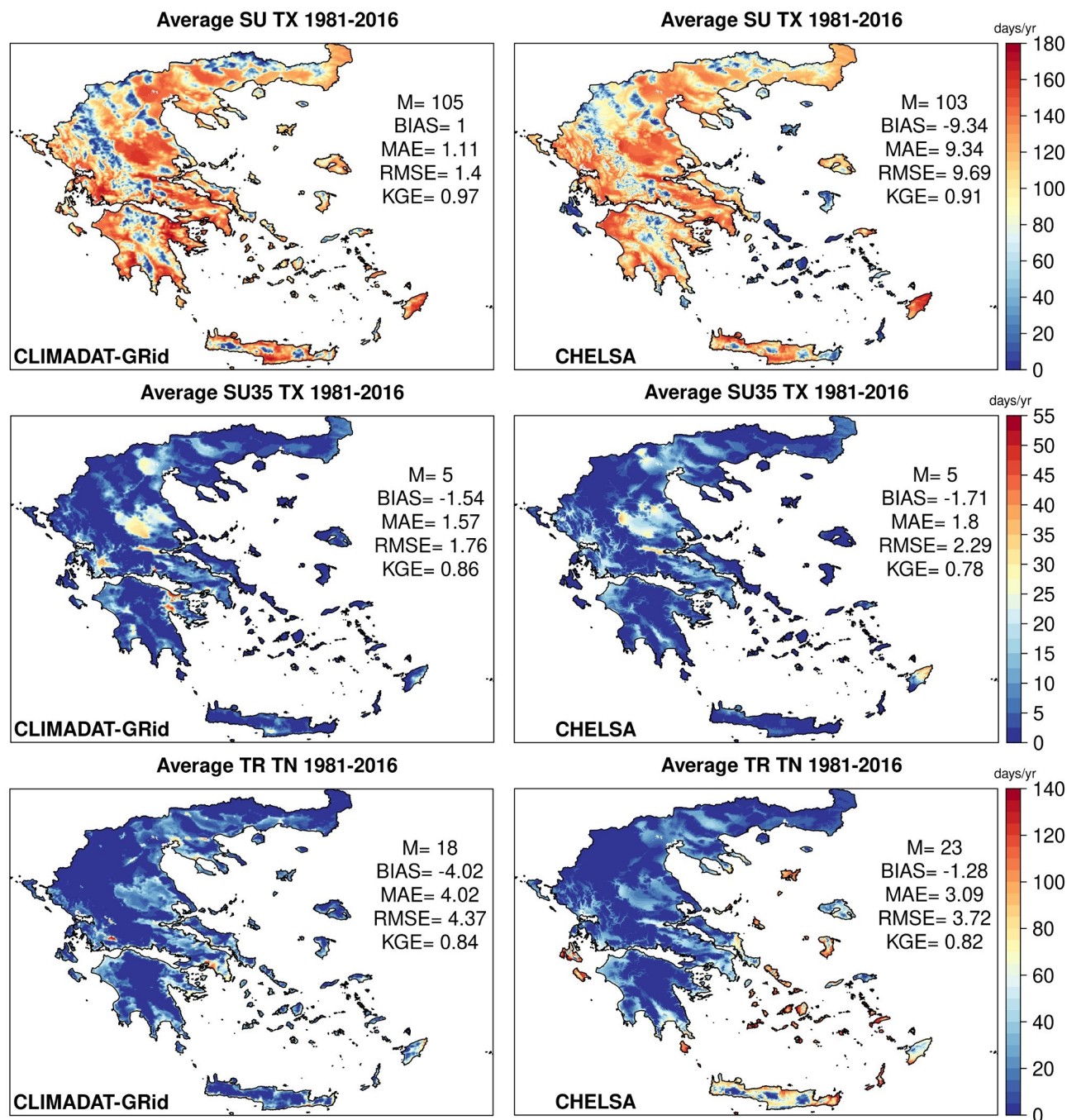

Figure 11. Average annual number of days *TX* > 25°C (SU), number of days *TX* > 35°C (SU35) and number of days *TN* > 20°C (TR) for the period 1981–2016 for CLIMADAT-GRid (left column) and CHELSA (right column). In each panel, M denotes the

spatial average over the grid points covering the area. In addition, the evaluation metrics between the stations and the data for the closest grid points to the stations locations are shown within each panel.

### 4.2.2 Precipitation results

Figure 12 presents the total annual and seasonal precipitation results averaged over the period 1981–2016 for both datasets. In general, CLIMADAT-GRid indicates higher precipitation values compared to CHELSA on both the annual and seasonal timescales (their relative differences are shown in Fig. S3 of the Supplementary Material). Both datasets capture the west-to-east precipitation gradient in Greece, with wetter conditions prevailing in the west and drier conditions in the east.

When evaluated against observations, CLIMADAT-GRid shows minimal biases. Specifically, the annual BIAS is −1.56%,
with seasonal biases ranging from −1% in DJF to −9.27% in JJA. CLIMADAT-GRid also maintains low MAE and RMSE values across all seasons, for example, annual MAE is 11.5 mm and RMSE is 15.17 mm, with high KGE values near 0.98, indicating strong agreement with observations.

In contrast, CHELSA demonstrates significant underestimations. The annual precipitation BIAS is −19%, and seasonal BIAS ranges from −11.97% in SON to −38.27% in JJA. The errors are also substantially larger, with annual MAE and
RMSE at 142.51 mm and 147.02 mm, respectively. Seasonal MAE and RMSE values are consistently higher than those of CLIMADAT-GRid, particularly during DJF, MAM and SON. Additionally, CHELSA's KGE values are lower across all seasons, with a peak of 0.82 in SON and a low of 0.65 in JJA, indicating comparatively reduced reliability in capturing observed precipitation patterns.

Regarding the number of wet days (RR1, Fig. 13 and Figure S4 of the Supplementary Material), both datasets demonstrate a
systematic overestimation relative to the observed spatial means. For CLIMADAT-GRid a positive BIAS of about 49 days/year is shown, while for CHELSA the bias reaches about 31 days/year. MAE and RMSE for CLIMADAT-GRid are about 49 and 50 days/year, respectively, with a negative KGE of −0.51, indicating poor agreement with observations despite its more pronounced orographic pattern. CHELSA performs somewhat better in this respect, with a lower MAE (31 days/year), RMSE (31 days/year), and a positive KGE of 0.31. This highly positive bias in the number of wet days has also
been found in other gridded products (e.g., IBERIA01), and it is a byproduct of the selected interpolation methods. One way to reduce the inflated number of wet days is to introduce a third term in the interpolation scheme of precipitation by interpolating the daily occurrence of rainfall (0 or 1 depending on whether $PR > 0.1$ mm) considering a threshold between 0.1 and 0.9 for assigning a wet day to a grid point (Cornes et. al., 2018; Varotsos et al., 2023a). For instance, if we assign a value of 0.2 for the wet days and multiply the interpolated fields with the daily precipitation product the average number of
wet days is reduced to 90 days/year with however increased underestimation in the annual and seasonal precipitation sums (not shown). For future studies utilizing the CLIMADAT-GRid precipitation dataset, a threshold of 2 mm/day could be considered when analyzing the number of wet days.

In terms of the number of days with daily precipitation equal to or greater than 10 mm (RR10, Fig. 13, Fig. S4 of the Supplementary Material), the two datasets display similar spatial distributions, with both indicating the highest frequencies

in western Greece and the lowest in the east. However, CLIMADAT-GRid performs better quantitatively, with a mean annual RR10 of 23 days and a bias of about -3 days/year, compared to CHELSA's 19 days/year and a larger negative bias of about −7 days/year. Additionally, CLIMADAT-GRid exhibits lower MAE and RMSE (3 days/year for both metrics, respectively), along with a higher KGE of 0.86, indicating close agreement with observations. CHELSA, in contrast, yields a higher MAE, RMSE (7 days/year for both metrics, respectively), and a lower KGE of 0.73, reinforcing the overall tendency

of CLIMADAT-GRid to more accurately represent moderate-to-heavy precipitation events.


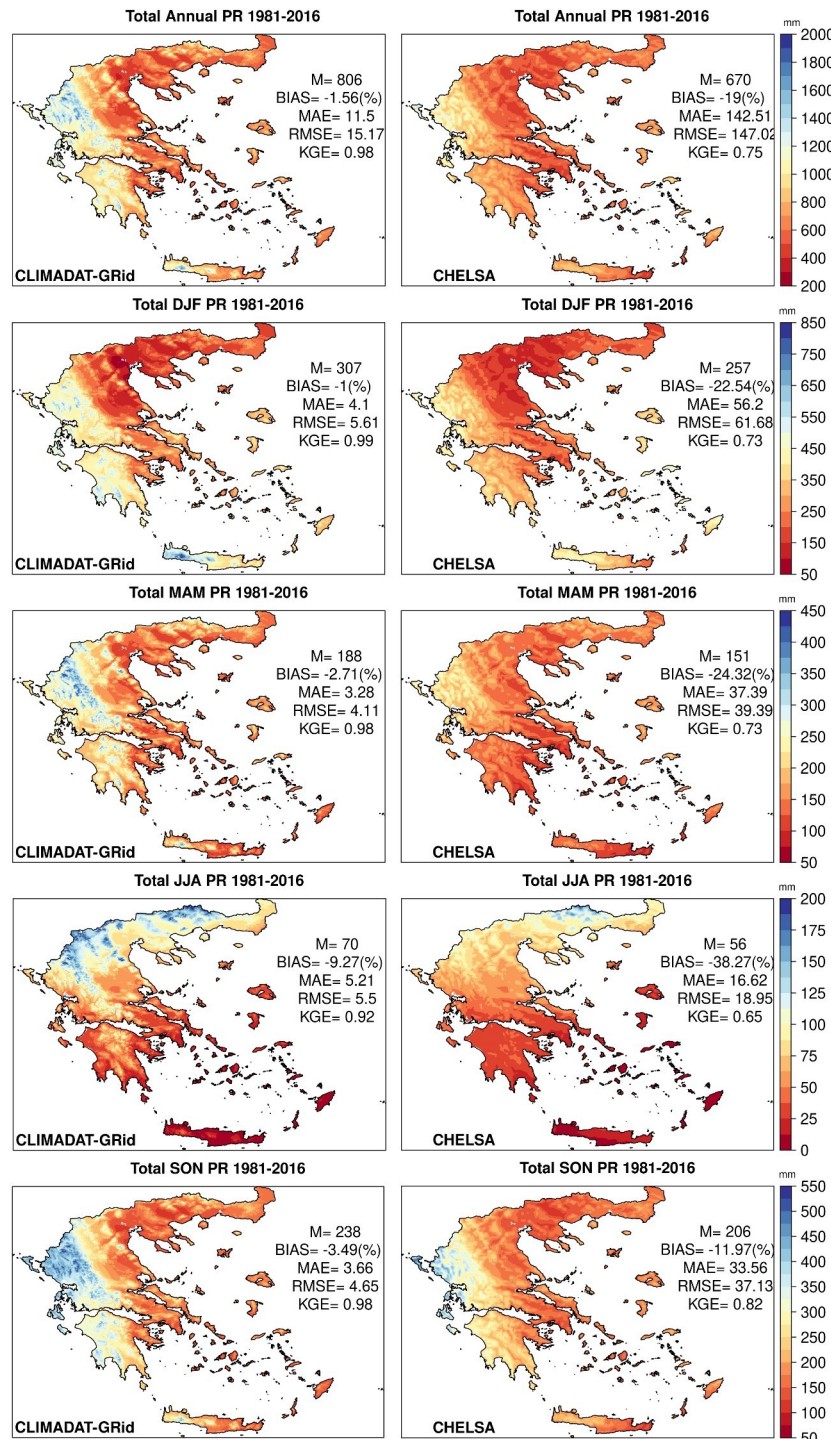

**Figure 12. Total annual and seasonal *PR* (winter (DJF), spring (MAM), summer (JJA) and autumn (SON)) for the period 1981–2016 for CLIMADAT-GRid (left column) and CHELSA (right column). In each panel, M denotes the spatial average over the grid**

points covering the area. In addition, the evaluation metrics between the stations and the data for the closest grid points to the stations locations are shown within each panel.

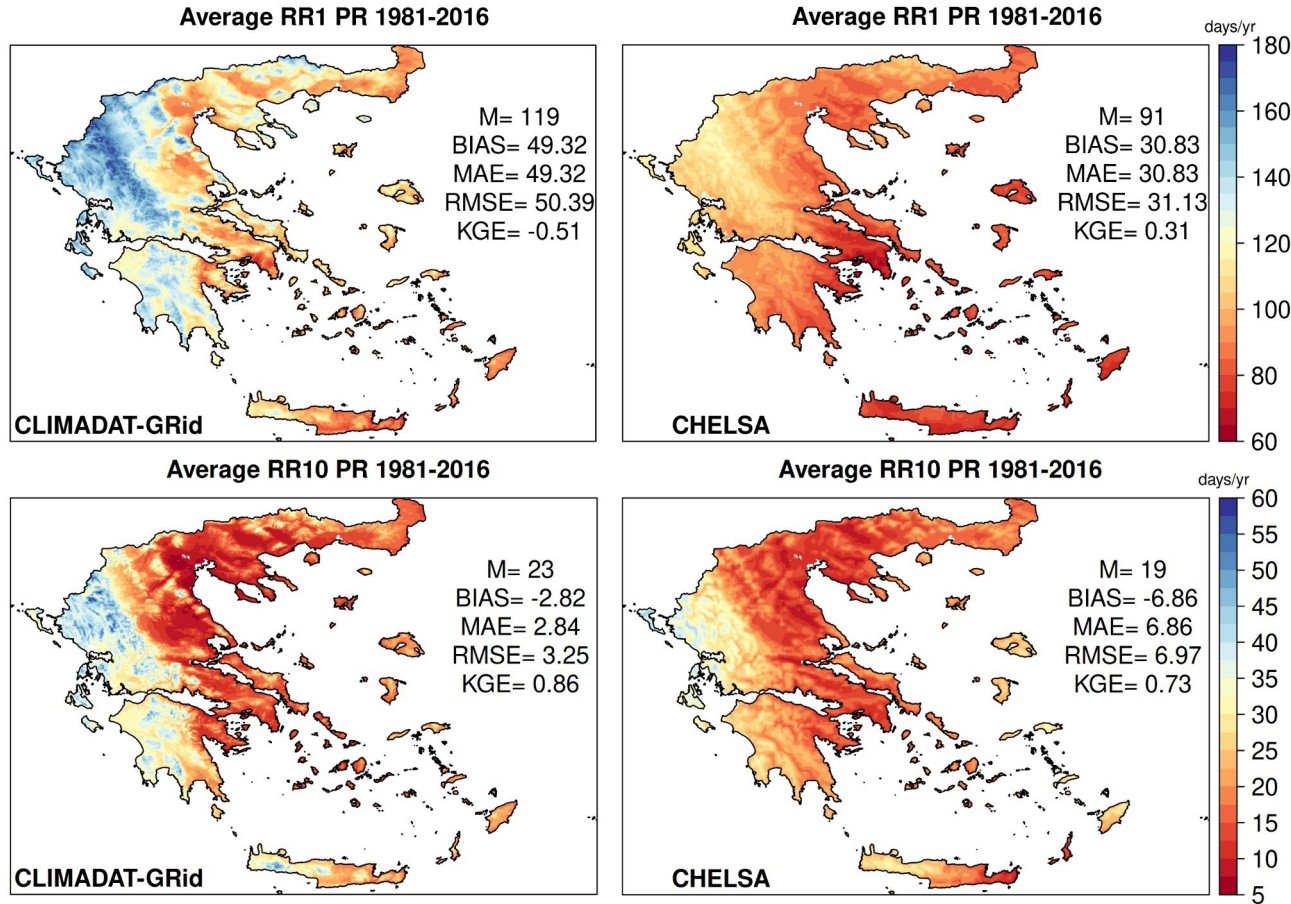

**Figure 13. Average annual number of days *PR* > 1 mm (RR1) and number of days *PR* >= 10mm (RR10) for the period 1981–2016 for CLIMADAT-GRid (left column) and CHELSA (right column). In each panel, M denotes the spatial average over the grid points covering the area. In addition, the evaluation metrics between the stations and the data for the closest grid points to the stations locations are shown within each panel.**

## 5 Data availability

The CLIMADAT-GRid dataset is freely available in the web repository Zenodo (https://doi.org/10.5281/zenodo.14637536) and cited as Varotsos et al. (2025). Moreover the dataset is available through http://ostria.meteo.noa.gr/repo/CLIMADAT_Grid/ (last access 19-12-2024). The NOAAN network measurements and the historical weather station data from the National Observatory of Athens in Thissio are available via the CLIMPACT data repository https://data.climpact.gr/en/dataset/497dc26d-45e0-4ad5-b8f3-5f8890f65129 and

https://data.climpact.gr/en/dataset/2f5bbe2a-7e27-40e7-9ff6-1dcc08c507fa, respectively (last access 20-3-2024). ERA5-land

were obtained from the Copernicus Climate Data Store (last access 17-4-2024) while CHELSA-W5 were obtained from https://data.isimip.org/10.48364/ISIMIP.836809.2 (last access 17-07-2024).

## 6 Conclusions

In this paper, we described the construction of CLIMADAT-GRid, a new publicly available 1 km x 1 km daily gridded climate dataset for Greece that focuses on temperatures and precipitation from 1981 to 2019. CLIMADAT-GRid is based on quality-controlled and homogenized daily temperature and precipitation data gathered from 122 and 312 stations, respectively. To produce the gridded fields, we evaluated four interpolation methods, Fixed Rank Kriging (FRK), Generalized Additive Models (GAM), Support Vector Machines (SVM), and K-Nearest Neighbors (KNN), using

independent station data for validation. FRK emerged as the most reliable method, demonstrating consistent performance across variables and time scales, particularly for precipitation. It also best captured spatial patterns, especially over the complex terrain of Greece. For temperatures, SVM and KNN performed well for maximum temperatures, while FRK was more consistent for mean and minimum temperatures. FRK was ultimately chosen as the method for constructing the CLIMADAT-GRid. In addition, to obtain the gridded temperature data the observations were blended with a high resolution

WRF simulation over Greece for the year 1999.

The comparison with the CHELSA-W5E5 dataset for the period 1981–2016 showed that CLIMADAT-GRid generally produced similar results for temperatures but captured spatial variability better with a closer agreement to observations on both the mean values and the extremes. For *TX*, both datasets showed similar temperatures, with CLIMADAT-GRid closely matching station data, while CHELSA consistently underestimating the observations by 0.5 to 0.7 °C. CLIMADAT-GRid

also better captured the temperature gradients in mountainous areas compared to CHELSA. Conversely, for *TN*, both datasets showed identical spatial means overall, with a tendency in CHELSA to overestimate observations. The differences between the datasets were most noticeable in the Ionian Sea islands, Crete, and the Aegean Sea islands, with CHELSA showing higher temperatures in these regions. Regarding the extremes, both datasets produced similar spatial means for the number of days with maximum temperatures above 25 °C and 35 °C, with CLIMADAT-GRid indicating the lowest biases

compared to the observations. However, CHELSA indicated a higher number of days with minimum temperatures above 20 °C compared to CLIMADAT-GRid. The spatial variability of these results is most noticeable in the Ionian and Aegean islands, with CLIMADAT-GRid effectively capturing hotspots. Overall, the study highlights the differences between CLIMADAT-GRid and CHELSA in capturing temperature variations in different regions, with CHELSA often underestimating or overestimating observations compared to CLIMADAT-GRid.

For precipitation, the comparison between CLIMADAT-GRid and CHELSA datasets for the annual and seasonal precipitation in Greece during the period 1981–2016 revealed that CLIMADAT-GRid generally indicated higher precipitation values compared to CHELSA. Both datasets capture the west-to-east gradient of precipitation in Greece, with

the differences being more pronounced in CLIMADAT-GRid, especially during the rainy season. When compared to observations, CLIMADAT-GRid had negligible biases, while CHELSA indicated relatively high biases ranging from 15-24 560 % depending on the season. Concerning the number of wet days, both datasets overestimate compared to observed spatial means, with CLIMADAT-GRid showing a more pronounced orographic pattern than CHELSA. Moreover, both datasets show similar results in the number of days with precipitation amounts equal to or higher than 10 mm, with CLIMADAT-GRid indicating higher values for this specific index in western Greece and better agreement with the observations.

In conclusion, CLIMADAT-GRid serves as a valuable resource for climate research in Greece, providing high-resolution 565 daily gridded datasets for temperatures and precipitation. Future work involves the construction of gridded datasets for other variables, such as relative humidity and wind speed, as well as extending the dataset to include more recent years.

**Author contribution**

KVV and CG conceptualized this study, GiKi, AK, IL, AS collected and provided the observational data, GeKa, VT, BP performed the quality control of the observational data, PP and MH performed the WRF simulations, KVV homogenized the 570 observational data, implemented the code to perform the interpolation and the analysis and created the dataset and figures of the paper; KVV designed and wrote the manuscript with contribution from CG, GeKa, AGK, MGG, AK and GiKi; all authors have read and approved the manuscript; financial support CG.

**Competing interests**

The authors declare that they have no conflict of interest.

**Acknowledgements**

The authors are grateful to the data providers of the Automatic Network and the Historical Weather Station of the National Observatory of Athens, the Hellenic National Meteorological Service and the General Secretariat for Natural Environment and Water of the Ministry of Environment and Energy.

**Financial support**

This study is part of the "High resolution gridded CLIMAte change DATasets for Greece, CLIMADAT-hub" project (https://www.climadathub.gr/). This project is carried out within the framework of the National Recovery and Resilience Plan Greece 2.0, funded by the European Union – NextGenerationEU (Implementation body: HFRI, Project ID: 15478)

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
