# Peer review of "CLIMADAT-GRid: A high-resolution daily gridded precipitation and temperature dataset for Greece"

_Earth System Science Data, 2025_

## Author Response (AR1)

**The manuscript presents observational gridded datasets over Greece, covering daily total precipitation and daily mean, maximum, and minimum temperatures. The authors have applied quality control and homogenization procedures to the input data. They also examined the use of different statistical methods for spatial interpolation. In addition, they incorporated numerical model output to address gaps in the observational network, which is relevant given the complex topography of the region. The datasets have been evaluated through cross-validation using independent observations and compared with existing gridded products available for the same area. The figures included in the paper are informative and clearly presented. The results support the conclusions drawn by the authors.**

**There are a few points that may require clarification or expansion. First, the manuscript does not include a sensitivity analysis regarding the use of WRF model output for a year other than 1999. While this analysis may not be essential, the authors could expand the discussion around lines 139–141. For example, they might consider whether a regional reanalysis product, such as CERRA, could have been used, or if WRF simulations were tested for other years. Second, certain methodological choices could be described in more detail. This is outlined in the comments below.**

**Overall recommendation: The study provides a useful dataset and analysis for the region. I recommend publication after the authors have addressed the comments that follow.**

Answer:

We sincerely appreciate the reviewer's thoughtful comments and valuable suggestions. We have carefully considered all the points raised and have addressed each of them thoroughly in the revised manuscript. We believe that these revisions have significantly improved the clarity, rigor, and overall quality of the work. A detailed point-by-point response is provided below, highlighting how each comment was incorporated or clarified.

Regarding the selection of the year for the WRF simulation. The specific selection of the simulation year is not of primary importance in this study, as the WRF model is used primarily as a physically based spatial interpolator. The model output is adjusted using observational data to account for seasonal and interannual variability. Therefore, the key requirement is that the WRF model provides a continuous and physically consistent representation of the temperature field across the region's complex terrain, a capability supported by the studies referenced in Section 2.3 of the revised manuscript. Thus the following lines have been added in section 2.3.

In particular, the **following lines** include studies that have implemented WRF in Greece as well as areas with similar topographic and climatic characteristics.

"WRF is widely used in both operational forecasting (Sofia et al., 2024; Patlakas et al., 2023) and scientific research (Pantillon et al., 2024; Patlakas et al., 2024; Politi et al., 2021; Stathopoulos et al., 2023; Otero-Casal et al., 2019). These studies provide comprehensive evaluations of the model's performance not only over the present study area but also in regions with similar topographic and climatic characteristics, demonstrating its reliability in representing climatological fields."

Moreover, the following lines have been added at the end of section 2.3.

"It should be noted here that the selection of the year of the WRF simulation is not of primary importance in this study, since it is used as a physically based spatial interpolator, as described in Section 3.2. Therefore, the key requirement is that the WRF model provides a continuous and physically consistent representation of the temperature field across the region's complex terrain, a capability supported by the aforementioned studies."

Regarding the comment on whether a regional reanalysis product such as CERRA could have been used, the methodology presented in this study indeed allows for that possibility, in several ways and for different purposes. For instance, if the goal is to develop a gridded dataset with a resolution similar to that of CERRA (5.5 km × 5.5 km), the WRF output could be directly replaced by CERRA. Alternatively, CERRA could be combined with WRF output to produce a statistically downscaled CERRA dataset, which can subsequently be bias-adjusted using observational data.

Specifically, in Step 1 of the methodology, the observations could be replaced by CERRA values at the closest grid points to the station locations. These values could then be used to perturb the WRF output, followed by applying the final step of the methodology to a 1 km regridded version of the CERRA dataset. Finally, the resulting dataset could be bias-adjusted by adding the interpolated mean monthly differences between the observations and the 1 km CERRA dataset (again, based on the closest grid points to the stations' locations).

To highlight the flexibility of the methodology the following lines have been added in the end of section **3.2 Spatiotemporal modeling for temperatures**

"The methodology presented in this study regarding the gridding of temperature data is flexible and allows for the integration of other regional datasets (e.g. the Copernicus regional reanalysis for Europe, CERRA) in multiple ways, depending on the objective. For example, if the aim is to develop a gridded dataset at a resolution similar to that of CERRA (5.5 km × 5.5 km), the WRF output could be replaced entirely with CERRA data. Alternatively, a combined approach could be employed, whereby CERRA is used in conjunction with WRF output to produce a statistically downscaled CERRA dataset, which can then be bias-adjusted using observational data.
More specifically, in the first step of the methodology, observational data could be substituted with CERRA values at the nearest grid points to the stations locations. These values would be used to perturb the WRF output, followed by application of the final step of the methodology to a 1 km regridded version of the CERRA dataset. The resulting high-resolution dataset could then be bias-adjusted by adding the interpolated mean monthly differences between the station observations and the corresponding values from the 1 km CERRA dataset."

**Comments:**

**1) Regarding the gridding of temperature data: It is likely that the station locations, your grid, and the CHELSA grid differ in elevation for the same geographic points. This is expected, but it is unclear how these differences were handled during the spatial analysis and subsequent comparisons. Did you interpolate all datasets onto a common grid before comparison? This point could be clarified in Sections 3.3 and 3.4. Also, discussing elevation differences may help**

**with the interpretation of results in Section 4.2.1. Please consider revising that section accordingly.**

Answer:

Both our gridded dataset (CLIMADAT-GRid) and the CHELSA dataset use the same underlying Digital Elevation Model (DEM), GMTED2010, as described in section 3.4. This ensures that elevation values in corresponding grid points are constant throughout the two gridded datasets. As a result, any systematic elevation differences exist exclusively between the station data and the gridded datasets, as station elevations may differ from the DEM-derived heights at the nearest grid points. As a result, we choose not to correct for elevation variations using a typical lapse-rate method. Nevertheless to further clarify that both gridded datasets are on the same grid the relative lines in section 3.4 have been modified **from:**

" CHELSA is a 1 km daily global land dataset for air temperatures, precipitation rates, and downwelling shortwave solar radiation for the period 1979–2016 and has been produced by spatially downscaling the 0.5° W5E5 dataset on an identical resolution grid as the one used in this study (GMTED2010)."

**to :**

" CHELSA is a global land dataset providing daily air temperature, precipitation, and downwelling shortwave solar radiation at a 1 km resolution for the period 1979–2016. It is produced by spatially downscaling the 0.5° W5E5 dataset onto a grid based on the GMTED2010 Digital Elevation Model, which is also used in this study. Notably, both the CLIMADAT-GRid and CHELSA are constructed using the same digital elevation model thus sharing the same grid while the shared elevation model ensures consistency in elevation values across corresponding grid points in the two datasets."

**2) The choice of FRK as the final spatial analysis method is only briefly mentioned in lines 289–291. This decision is important and could be stated earlier and more clearly. For example, it could be introduced in the abstract (e.g., after "against withheld observational data," add a sentence about the method used). Additionally, you could move the relevant lines to the beginning of Section 4.1, rather than introducing FRK in the section discussing temperature results. Consider also whether the conclusion should briefly mention that FRK performed best among the methods evaluated. It may also be useful to explain why a single method (FRK) was chosen for both temperature and precipitation, despite indications that SVM performed well for precipitation. A short explanation of the reasoning behind this choice could be helpful.**

Answer:

Following the reviewer's suggestion, the abstract has been modified to include the method selected.

In particular, the abstract has been modified **from:**

[revised manuscript text omitted]

In addition to the above temperature data, daily precipitation observations were collected from 190 stations operated by the General Secretariat for Natural Environment and Water of the Ministry of Environment and Energy, covering the period 1981–2019. Combined with precipitation records from HNMS and NOAAN, this results in a total of 312 stations (Fig. 1b), with station altitudes ranging from sea level to 1130 m a.s.l. Only stations with less than 10% missing data annually were considered. According to the data providers, daily precipitation data were collected over a 24-period from 08:00 to 08:00 UTC for the HNMS, NOA and the stations provided by the General Secretariat for Natural Environment and Water of the Ministry of Environment and Energy. Regarding the NOAAN stations, daily precipitation data were collected over a 24-period from 00:00 to 24:00 UTC."

**5. Line 165: Consider whether this line should be part of the previous paragraph, as the new line may not be necessary.**

Answer:

The reviewer's suggestion has been implemented.

**New references included in the revised version of the manuscript and in the response letter.**

Founda, D., Katavoutas, G., Pierros, F., and Mihalopoulos, N.: Centennial changes in heat waves characteristics in Athens (Greece) from multiple definitions based on climatic and bioclimatic indices, Global and Planetary Change, 212, 103807, https://doi.org/10.1016/j.gloplacha.2022.103807, 2022.

Pantillon, F., Davolio, S., Avolio, E., Calvo-Sancho, C., Carrió, D. S., Dafis, S., Gentile, E. S., Gonzalez-Aleman, J. J., Gray, S., Miglietta, M. M., Patlakas, P., Pytharoulis, I., Ricard, D., Ricchi, A., Sanchez, C., and Flaounas, E.: The crucial representation of deep convection for the

cyclogenesis of Medicane Ianos, Weather and Climate Dynamics, 5, 1187–1205, https://doi.org/10.5194/wcd-5-1187-2024, 2024.

Patlakas, P., Chaniotis, I., Hatzaki, M., Kouroutzoglou, J., and Flocas, H. A.: The eastern Mediterranean extreme snowfall of January 2022: synoptic analysis and impact of sea-surface temperature, Weather, 79, 25–33, https://doi.org/10.1002/wea.4397, 2024.

Politi, N., Vlachogiannis, D., Sfetsos, A., and Nastos, P. T.: High-resolution dynamical downscaling of ERA-Interim temperature and precipitation using WRF model for Greece, Climate Dynamics, 57, 799–825, https://doi.org/10.1007/s00382-021-05741-9, 2021.

**Reviewer 2**

**MAJOR COMMENTS**

**The manuscript presents a comprehensive dataset derived from regional climate downscaling addressed to the Greek territory using advanced machine learning techniques. The primary objectives are to enhance the spatial resolution over a 39-year period and to improve the representation of daily temperature and precipitation fields across the complex geography of continental Greece and its islands. The authors employ a hybrid approach combining geostatistical interpolation and statistical downscaling methods with atmospheric modelling, validated against observational datasets. Four methods (FRK, GAM, KNN, and SVM) were evaluated, with FRK ultimately chosen. Evaluation against CHELSA-W5E5 and withheld station data supports the improved spatial accuracy and bias reduction of CLIMADAT-GRid, especially in mountainous regions. The validation strategy, multiple error metrics, and comparison with an established product (CHELSA-W5E5) strengthen the study's robustness. The outcomes suggest improved accuracy in temperature and precipitation at regional scales, supported by the analysis of suitable climate indicators. However, while the objectives are met mainly, certain aspects require further clarification to substantiate the claims thoroughly.**

Answer:

We sincerely appreciate the reviewer's thoughtful comments and valuable suggestions. We have carefully considered all the points raised and have addressed each of them thoroughly in the revised manuscript. We believe that these revisions have significantly improved the clarity, rigor, and overall quality of the work. A detailed point-by-point response is provided below, highlighting how each comment was incorporated or clarified.

**1) The introduction chapter might discuss similar datasets produced for the same purpose. This would help clarify the expectations surrounding this exercise, including its benefits, drawbacks, and potential challenges. The datasets E-OBS and IBERIA01 referenced by the authors in Chapter 3 could serve as a starting point.**

Answer:

Following the reviewer's suggestion, the specific section in the introduction has been modified in the revised manuscript **from:**

[revised manuscript text omitted]

**2) In the subsequent chapter, the datasets are delineated without any preceding explanation or introduction regarding their presentation, which may lead to confusion for the readers. For example, WRF parachutes in Section 2.3 without any justification or preangle (abstract is not**

**part of the manuscript), leading readers to assume that the model will be employed for temperature and precipitation analysis. I recommend incorporating an introductory paragraph between items 2 and 2.1 to bridge this gap, as was implemented in Chapter 3.**

Answer:

Following the reviewer's suggestion, the following lines have been added between 2 and 2.1 in the revised manuscript.

"In this section, the datasets utilized in the analysis are presented. Subsection 2.1 summarizes the daily observational data, including maximum (TX), minimum (TN), and average (TG) temperatures, as well as daily precipitation (PR). Subsection 2.2 outlines the procedures applied for quality control, gap filling, and homogenization of the datasets. Subsection 2.3 describes the Weather Research and Forecasting (WRF) model simulation, whose output is blended with the available temperature observational data using gridding techniques, as detailed in Section 3. This approach was preferred over relying solely on observational data due to the sparse spatial coverage of in situ measurements, especially at higher altitudes (above 1000 m) as presented in Subsection 2.1."

**3) Continuing with the discussion on WRF, only during Section 3.2 (the second section before the results), the readers are informed that the atmospheric model was utilised exclusively for the temperature field, which may reduce the audience rejection regarding precipitation. The decision to employ solely one year to represent the overall study period is somewhat contentious, as a significant amount of variability is forfeited. Nevertheless, this approach is permissible, given that the model ultimately functions as a spatial interpolator driven by physical laws, subsequently manipulated to incorporate the seasonal and interannual variations delineated by the observational data. It may be beneficial for the authors to include a map illustrating the participation of the observational data at each grid point, as it could mitigate the discussion concerning the employment of the atmospheric model to cover regions lacking a station. Furthermore, this addition would facilitate the analysis of the specific areas to which the results can be attributed through the model. It would be essential to formally present the domain's limits, since they may circumstantially restrict the representation of some large-scale atmospheric phenomena transiting from the boundary condition. Furthermore, the authors reference several studies on WRF applications in Greece but do not elaborate on their findings or the model calibration expressed by the selected set of physical parameterisations. This potentially makes the WRF application quite questionable for these purposes.**

Answer:

We appreciate the reviewer's constructive suggestions. Given the 1 km resolution of the WRF model, it is indeed the case that the vast majority of 1 km × 1 km grid cells lack direct in situ observations. This spatial sparsity is a primary reason for our reliance on model-derived output, which offers a continuous and physically consistent representation of the temperature field. The absence of observational stations, particularly in areas above 1000 meters, further justifies our methodological choice, as the model helps fill observational gaps in complex terrain. Please see our response in the previous comment.

Regarding the reviewer's comment on the model extent, we agree that this clarification enhances the transparency of our modeling framework. Accordingly, we have included the following figure

(Figure 2, new) in the revised manuscript that clearly depicts the model domain and its geographical boundaries. This addition enables the reader to assess the potential influence of boundary conditions on the study area and better understand the spatial context of the simulations.

[Figure]

Figure 2: WRF-ARW model domains

Finally, in response to the comment on model performance, we have added references to studies where similar WRF setups have been used and evaluated. These studies report robust performance in comparable climatic and topographic settings. A clarifying sentence has also been included to highlight the prior validation of the model configuration used in our study.

Therefore, section 2.3 has been modified **from:**

"For the atmospheric simulations, the Advanced Weather Research and Forecasting Model (WRF-ARW) version 4.1.3 (Powers et al., 2017; Skamarock et al., 2008; Skamarock et al., 2019) was employed. WRF-ARW serves as a limited-area atmospheric model, utilized for both operational forecasting (Sofia et al., 2024; Patlakas et al., 2023) and scientific research (Stathopoulos et al., 2023; Otero-Casal et al., 2019). It is based on a fully compressible, non-hydrostatic dynamic core. On the vertical plane it has terrain-following, mass-based, hybrid sigma-pressure vertical coordinates based on dry hydrostatic pressure, with vertical grid stretching permitted while for the horizontal grid, the Arakawa C-grid staggering is employed.
In this analysis, the WRF model was configured to run with three two-way nested grids. The coarser one has a resolution of 9 km, covering a large area that includes parts of North Africa and Central Europe. The inner grids are focused on the Eastern Mediterranean and Greece, with spatial resolutions of 3 km and 1 km, respectively. Vertically, the model consists of 48 layers."

**to :**
"For the atmospheric simulations, the Advanced Weather Research and Forecasting Model (WRF-ARW), version 4.1.3 (Skamarock et al., 2019), was employed. WRF-ARW is a limited-area atmospheric model based on a fully compressible, non-hydrostatic dynamic core. Vertically, it

utilizes terrain-following, mass-based hybrid sigma-pressure coordinates based on dry hydrostatic pressure, with support for vertical grid stretching. Horizontally, the model applies an Arakawa C-grid staggering.

WRF is widely used in both operational forecasting (Sofia et al., 2024; Patlakas et al., 2023) and scientific research (Pantillon et al., 2024; Patlakas et al., 2024; Politi et al., 2021; Stathopoulos et al., 2023; Otero-Casal et al., 2019). These studies include comprehensive evaluations of the model's performance not only over the present study area but also in regions with similar topographic and climatic characteristics, demonstrating its reliability in representing climatological fields. In this analysis, the WRF model was configured with three two-way nested grids to adequately capture both regional and local-scale processes. The inner grids are focused on the Eastern Mediterranean and Greece, with spatial resolutions of 3 km and 1 km, respectively (Figure 2). Vertically, the model consists of 48 layers."

**4) The description of geostatistical interpolation and ML methods, lacking detail and overly restricted to their respective R packages citation, is both awkward and limiting (Sect. 3.1 and 3.2). This narrow focus obscures the wide variety of options and parameters defined in each method and, as a result, hinders and prevents the proper reproducibility of the study. Thus, presenting these details in the manuscript is crucial.**

Answer:

Following the reviewer's suggestion, the following lines have been modified in section 3.1 of the revised manuscript **from:**

"The following approaches were examined to calculate the monthly precipitation fields: (i) a "fixed rank Kriging approach" (hereafter FRK, Nychka et al., 2015), (ii) generalized additive models (hereafter GAM, Wood, 2017) and (iii) two ML algorithms namely k-nearest neighbors (hereafter KNN) and support vector machines (hereafter, SVM). The analysis is performed using the R Language and Environment for Statistical Computing (R core team 2024) and the packages LatticeKrig (Nychka et al., 2019) for FRK, mgcv (Wood and Wood, 2015) for GAM, while the two ML algorithms are tuned using the caret R package (Kuhn, 2008)."

**to :**

"The following approaches were examined to calculate the monthly precipitation fields:
(i) a "Fixed Rank Kriging approach" (FRK). FRK is a geostatistical interpolation technique that approximates a spatial field using a low-rank representation of the underlying spatial process. It models the spatial covariance structure through a set of basis functions, allowing efficient estimation even with large datasets (Nychka et al., 2015). In this study, FRK is implemented using the LatticeKrig package (Nychka et al., 2019) in R (R Core Team, 2024), where the model parameters, including variance components and spatial range parameters, are estimated using maximum likelihood estimation.
(ii) Generalized Additive Models (hereafter GAM) are a semi-parametric extension of Generalized Linear Models that assume the underlying relationships are additive and smooth. Their primary strength lies in their ability to capture highly non-linear and non-monotonic relationships between the response variable and explanatory variables (Wood, 2017). In this study, monthly precipitation sums are modeled as smooth functions of longitude, latitude and elevation using thin plate

regression splines, with smoothing parameters estimated using restricted maximum likelihood (mgcv R package, Wood and Wood, 2015).

(iii) two ML algorithms namely K-Nearest Neighbors (KNN) and Support Vector Machines (based on an exponential radial basis function, SVM). KNN estimates the value of an unknown data point by identifying its k closest neighbors in the spatial dataset, where k is a user-defined hyperparameter (Nwaila et al., 2024). The predicted value is computed as a weighted average of these neighbors' values, with the weights typically based on the distance to the target point, i.e. closer neighbors have greater influence. In this study, k ranges from 2 to 30 in increments of 1. SVM is a ML algorithm, effective in capturing non-linear spatial trends, which seeks a function that predicts the value of an unknown data point, while balancing accuracy and model complexity (Bonsoms and Ninyerola, 2024). The complexity is regulated by the cost parameter C (tested values: 0.1, 1, 5, 10), while the smoothness of the kernel is governed by *sigma* (tested values: 0.01, 0.025, 0.05, 0.075, 0.1). For both algorithms, optimal hyperparameters (k for KNN, and C and sigma for SVM) are selected using tenfold cross-validation combined with grid search, using the caret package in R (Kuhn, 2008; R Core Team, 2024)."

**5) Concerning the metrics employed in the evaluation, specifically RMSE, MAE, and KGE, it is noteworthy that the most fundamental among them, Bias, was not considered. Relying solely on RMSE and MAE does not allow assessing whether the method underestimates or overestimates the observed values. This significantly impacts the analysis of temperature fields, particularly precipitation fields. Therefore, including this metric is also mandatory.**

Answer:

Following the reviewer's suggestion BIAS values are also reported in Tables 2 and 3 of the revised manuscript. Moreover, the discussion in sections 4.1.1 and 4.1.2 have been modified to include the BIAS values. In particular, for precipitation in section 4.1.1 the following lines have been modified **from:**

[revised manuscript text omitted]
 | 0.06 | 0.48 | 0.38 | 0.99 | 0.45 | 3.85 | 0.54 | 0.96 | 0.74 | 0.88 | 0.77 | 0.94 | 0.62 | 0.79 | 0.68 | 0.95 |

**6) The results chapter is well-structured, and its figures and tables are clear, readable, and sufficient to justify the main findings. Nevertheless, it is also true that omitting some information and making certain choices in preparing the outputs reduces their respective impacts. For example, why was Figure 4 presented only for 2016 and not for the 10 years of validation (2010-2019)? Besides lacking a direct relation to Table 3, this leads readers to believe that the differences pointed out by the authors are limited and specific to that year and may not be as pronounced in an overall analysis. Furthermore, including a fifth panel (e) with the map that highlights only the station locations using colours representing the observed values on the same scale as precipitation would significantly enhance this analysis and assist in determining which method was definitively superior. Keep in mind that spatial variability does not imply a better result.**

Answer:

Following the reviewer's suggestion, a new Figure 4 has been added in the revised manuscript replacing Figure 4 from the first version of the manuscript. In addition, the text in section 4.1.1 has been modified **from:**

"In terms of spatial distribution, the west to east gradient of precipitation in Greece, which exhibits the largest and lowest amounts of yearly precipitation (Gofa et al., 2019), is captured by all approaches. However, as Fig. 4 for the year 2016 illustrates, GAM, KNN and SVM demonstrate reduced spatial variability in regions with significant altitude differences in both the northeastern

and southern regions of Greece (e.g. Crete). This spatial variability is also evident in other years examined within the 2010–2019 period (not shown).”

**to:**

“ When all available stations are used for the interpolation for the period 2010-2019 (Fig. 4), the results indicate that the west to east gradient of precipitation in Greece, which exhibits the largest and lowest amounts of yearly precipitation (Gofa et al., 2019) is captured by all approaches. However, as it is evident from Fig. 4, FRK maintains strong predictive skill when applied to the full set of available stations, with only a modest increase in error, reflecting its ability to handle spatial heterogeneity and non-stationarity common in precipitation fields. GAM error metrics increase somewhat on the full dataset, however, retaining a reasonable predictive skill, indicating its capacity to capture important spatial patterns. For KNN, an opposite sign BIAS is evident when compared to the results of Table 2 indicating that the method is strongly dependent on the proximity to known data. As a result, the method exhibits poorer extrapolation ability than FRK and GAM. In contrast, SVM shows a significant decline in performance when applied to the full station network, as it lacks explicit modeling of spatial structures despite its capability to capture complex nonlinear relationships (Heinke et al. 2023), resulting in higher errors and biases across diverse climatic and geographic regions. This behavior is highlighted in the mountainous area in Crete where total annual precipitation is much lower than the other methods.

Overall, this comparison highlights that FRK is better suited for robust precipitation interpolation across regions with complex topography.”

[Figure]

**Figure 4. Spatial distribution of total annual precipitation for the period 2010–2019, as estimated by the different interpolation methods (panels a–d) and observed data (panel e). Each panel includes the spatial average (M) calculated over all grid points (or over all stations in panel e). For panels a–d, the relative BIAS, MAE (in mm), RMSE (in mm), and KGE are provided, based on comparisons between the interpolated values at the nearest grid points and the corresponding station observations.**

In addition, the following lines 285-289 in subsection 4.1.2 have been modified **from:**

"In terms of spatial distribution, similar patterns to those observed for precipitation are identified. In particular GAM, KNN and SVM demonstrate reduced spatial variability between the mountainous areas of Greece and the lower altitude surrounding areas indicating limited ability to accurately represent temperature gradients influenced by topography (not shown)."

**to:**

" In terms of spatial distribution, similar patterns to those observed for precipitation are identified. In particular GAM, KNN and SVM demonstrate limited ability to accurately represent temperature gradients influenced by topography (not shown)."

**7) Another relevant point regards the sentence "Despite this, CHELSA underestimates the observed number of SU by about 10 days/yr, while CLIMADAT-GRid closely aligns with the observed values." (L319-320). Which results substantiate this conclusion? It may be pertinent to present and discuss the evaluation metrics (Bias, RMSE, MAE, KGE) for both CLIMADAT-GRid and CHELSA throughout the entire study period. Only a brief and vague text in L350-354 may not be sufficient.**

Answer:

In the first version of the manuscript in each panel of the Figures 8-12 the following values were reported within the panels: M which denoted the spatial average over the grid points covering the area, M.o which denoted the station mean values and M.c which denoted the mean values for the closest grid points to the stations locations. Therefore, the difference between the interpolated values at the nearest grid points (M.c) and the corresponding station observations (M.o) could be calculated. Nevertheless, in the revised version of the manuscript all the evaluation metrics are presented within each panel in addition to TG (the new figures can be found at the end of this response letter). Moreover, the differences between CLIMADAT-GRid and CHELSA can be found in the supplementary material (the new figures can be found at the end of this response letter). Consequently, the discussion in section 4.2 has been modified to incorporate all the changes. In particular, the following lines have been modified **from:**

[revised manuscript text omitted]

**8) Finally, the conclusions chapter effectively fulfills its intended role, although it does not provide commentary on an essential aspect of the work regarding the various geostatistical and machine learning techniques employed in developing the temperature and precipitation datasets. This oversight may be attributed to the insufficient detail in the preceding chapters. Incorporating these elements, whether in the methodology or the conclusions, would substantially enhance the manuscript's value. Nonetheless, it is essential to underscore that the analyses provided are devoid of any fallacies or significant flaws and, in any case, compromise the integrity of the study. It is simply a matter of refinement.**

Answer:

Following the reviewer's suggestion the following lines have been added in the Conclusions section:

"To produce the gridded fields, we evaluated four interpolation methods, Fixed Rank Kriging (FRK), Generalized Additive Models (GAM), Support Vector Machines (SVM), and K-Nearest Neighbors (KNN), using independent station data for validation. FRK emerged as the most reliable method, demonstrating consistent performance across variables and time scales, particularly for precipitation. It also best captured spatial patterns, especially over the complex terrain of Greece. For temperatures, SVM and KNN performed well for maximum temperatures, while FRK was more consistent for mean and minimum temperatures. FRK was ultimately chosen as the method for constructing the CLIMADAT-Grid."

**Given the innovative approach and the potential contributions to regional climate, I recommend acceptance upon a comprehensive review of major comments. The paper presents a high-quality, methodologically sound dataset likely to be of great use in regional climate research, impact modelling, and policy work in Greece. The authors must tackle the previously mentioned concerns to enhance their transparency, reproducibility, and broader significance. Addressing these issues will fortify the manuscript and increase its contribution to the scientific community.**

Answer:

We thank the reviewer for their positive evaluation and constructive feedback. We have carefully addressed all issues raised, improving the manuscript's transparency, reproducibility, and overall clarity.

**MINOR COMMENTS**
**L23, L26  CHELSA still CHELSA-W5E5 up to this point.**

Corrected

**L31 The phrase "… are becoming…" requires modification. This citation originates from 2012. Currently, it represents a prevailing reality.**

Modified

**L32-33 Add a comma in "(Herrera et al. 2012)".**

Corrected

**L55 Remove the E-OBS citation that was previously introduced in L41.**

Corrected

**L63 Citation missing for "MeteoSerbia1km".**

Corrected

**L73 The acronym "CLIMADAT-GRid" is used without prior definition. Please define it upon first use.**

CLIMADAT-GRid is a designated name and not an acronym

**Fig 1 Figures 1 and 2 are redundant; only Figure 2 is sufficient if it replaces Figure 1. Furthermore, the blue colouration on terrain elevation maps is typically attributed to regions situated below the mean sea level (h<0). Consequently, it is advisable to redefine the scale to initiate with green tones.**

Following the reviewer's suggestion, the following figure has been added in the revised manuscript which replaces Figures 1 and 2 of the first version of the manuscript.

[Figure]

*Figure 1. Locations of meteorological stations used for (a) temperature and (b) precipitation measurements, including both the stations used in the interpolation and the withheld stations used for evaluation. The background shows elevation data from the Global Multi-resolution Terrain Elevation Data 2010 (GMTED2010).*

**L120 Citing *Skamarock et al. (2019)* may be sufficient.**

The reviewer's suggestion has been implemented.

**L136 Remove the term "*approximately*" once ERA5 has a precise horizontal resolution of 0.25º. In this case, the approximation regards the resolution in km, which varies from 25 to 31 km, roughly estimated at 28 km.**

The reviewer's suggestion has been implemented.

**L137 Following the standard presented in the manuscript, replace "*USGS (United States Geological Survey) (Slater et al., 2011)*" with "*United States Geological Survey (USGS, Slater et al., 2011)*".**

The reviewer's suggestion has been implemented.

**L137 Following the standard presented in the manuscript, replace "*CORINE (Coordination of Information on the Environment) database (2010)*" with "*Coordination of Information on the Environment (CORINE, CLMS 2018)*" if the authors have used the latest version (https://doi.org/10.2909/960998c1-1870-4e82-8051-6485205ebbac). Additionally, this citation may be included in the references.**

The reviewer's suggestion has been implemented.

**L153 The acronym "GMTED2010" is used without prior definition. Please define it upon first use (Figure's caption doesn't count).**

The reviewers' suggestion has been implemented.

**L202 Remove the comma in "*Papa and Koutroulis (2025,)*".**

The reviewers' suggestion has been implemented.

**L205 Replace "*Climate Change Detection and Indices (ETCCDI) (Zhang et al. 2011).*" with "*Climate Change Detection and Indices (ETCCDI, Zhang et al., 2011).*".**

The reviewers' suggestion has been implemented.

**Fig 2 In addition to the aforementioned comments regarding Figure 1, it is advisable to change the colour used for the markers on the evaluation stations, as they tend to blend with the background.**

The reviewers' suggestion has been implemented.

**L217 Replace "*The values of the root mean square error (RMSE), the mean absolute error (MAE) and the KGEs…*" with "*The values of RMSE, MAE and the KGEs…*", once they were defined previously.**

The reviewers' suggestion has been implemented.

**Fig 8,9,11 Since the objective of these figures is to compare the temperature and precipitation fields of two different datasets, wouldn't it be better to present the difference between them instead of the entire fields? This way, the differences pointed out by the authors would be clearer. Furthermore, it would enable the presentation of TN, TG, and TX in the same figure without losing quality. That is, Figures 8 and 9 would be merged, with the addition of TG, which was omitted without explanation.**

We have retained the original structure for presenting the comparison between the two datasets in the revised manuscript. This approach not only facilitates the comparison itself but also serves to illustrate how temperature and precipitation fields are distributed across the complex terrain of Greece, which is a key objective of the study. However, to address the reviewer's concern, we have included the differences between the two datasets for the annual and seasonal means, as well as for the climate indices, in the Supplementary Material. Additionally, a new figure related to TG (new Figure 9) has been incorporated and discussed in the revised manuscript. Please also refer to our response to Comment 7 in the list of major comments for further clarification.

**General The recursive use of "*hereafter*" is inappropriate in most instances. Typically, this expression is employed to redefine a name or acronym. Only in L199 does it appear to be correctly utilised to redefine the acronym CHELSA-W5E5 as CHELSA.**

The reviewers' suggestion has been implemented.

**General In the scientific literature on climate and meteorology, the prevailing terminology for the temporal aggregation of precipitation over a day is "*daily accumulated precipitation*" or simply "*daily precipitation*". Although the procedure is referred to as the precipitation sum, its use can lead to different interpretations.**

The reviewers' suggestion has been implemented. Daily precipitation is used in the revised manuscript.

[Figure]

*Figure 8. Average annual and seasonal TX (winter (DJF), spring (MAM), summer (JJA) and autumn (SON)) for the period 1981–2016 for CLIMADAT-GRid (left column) and CHELSA (right column). In each panel, M denotes the spatial average over the grid points covering the area. In addition, the evaluation metrics between the stations and the data for the closest grid points to the stations locations are shown within each panel.*

[Figure]

Figure 9. Average annual and seasonal TG (winter (DJF), spring (MAM), summer (JJA) and autumn (SON)) for the period 1981–2016 for CLIMADAT-GRid (left column) and CHELSA (right column). In each panel, M denotes the spatial average over the grid points covering the area. In addition, the evaluation metrics between the stations and the data for the closest grid points to the stations locations are shown within each panel.

[Figure]

*Figure 10. Average annual and seasonal TN (winter (DJF), spring (MAM), summer (JJA) and autumn (SON)) for the period 1981–2016 for CLIMADAT-GRid (left column) and CHELSA (right column). In each panel, M denotes the spatial average over the grid points covering the area. In addition, the evaluation metrics between the stations and the data for the closest grid points to the stations locations are shown within each panel.*

[Figure]

*Figure 11. Average annual number of days TX > 25°C (SU), number of days TX > 35°C (SU35) and number of days TN > 20°C (TR) for the period 1981–2016 for CLIMADAT-GRid (left column) and CHELSA (right column). In each panel, M denotes the spatial average over the grid points covering the area. In addition, the evaluation metrics between the stations and the data for the closest grid points to the stations locations are shown within each panel.*

[Figure]

*Figure 12. Total annual and seasonal PR (winter (DJF), spring (MAM), summer (JJA) and autumn (SON)) for the period 1981–2016 for CLIMADAT-GRid (left column) and CHELSA (right column). In each panel, M denotes the spatial average over the grid points covering the area. In addition, the evaluation metrics between the stations and the data for the closest grid points to the stations locations are shown within each panel.*

[Figure]

*Figure 13. Average annual number of days PR > 1 mm (RR1) and number of days PR >= 10mm (R10mm) for the period 1981–2016 for CLIMADAT-GRid (left column) and CHELSA (right column). In each panel, M denotes the spatial average over the grid points covering the area. In addition, the evaluation metrics between the stations and the data for the closest grid points to the stations locations are shown within each panel.*

**Supplementary material figures**

[Figure]

*Fig S1. Average annual and Seasonal differences for TX, TG and TN between CLIMADATGRid and CHELSA for the period 1981-2016.*

[Figure]

Fig S2. Average annual differences for SU, SU35 and TR between CLIMADATGRid and CHELSA for the period 1981-2016.

[Figure]

*Fig S3. Average annual and seasonal relative differences in precipitation between CLIMADAT-Grid and CHELSA for the period 1981–2016.*

[Figure]

*Fig S4. Average annual differences for RR1 and RR10 between CLIMADATGRid and CHELSA for the period 1981-2016.*

**New references included in the revised version of the manuscript and in the response letter.**

Serrano-Notivoli, R., Beguería, S., Saz, M. Á., Longares, L. A., and de Luis, M.: SPREAD: a high-resolution daily gridded precipitation dataset for Spain–an extreme events frequency and intensity overview, Earth System Science Data, 9, 721–738, https://doi.org/10.5194/essd-9-721-2017, 2017.

Serrano-Notivoli, R., Beguería, S., and de Luis, M.: STEAD: a high-resolution daily gridded temperature dataset for Spain, Earth System Science Data, 11, 1171–1188. https://doi.org/10.5194/essd-11-1171-2019, 2019.

Serrano-Notivoli, R., Saz, M. Á., Longares, L. A., and de Luis, M.: SiCLIMA: High-resolution hydroclimate and temperature dataset for Aragón (northeast Spain). Data in Brief, 56, 110876, https://doi.org/10.1016/j.dib.2024.110876, 2024.

Fonseca, A. R., Santos, J. A.: High-Resolution Temperature Datasets in Portugal from a Geostatistical Approach: Variability and Extremes. Journal of Applied Meteorology and Climatology, 57, 627–644, https://doi.org/10.1175/JAMC-D-17-0215.1, 2018.

Krähenmann, S., Walter, A., Brienen, S., Imbery, F., and Matzarakis, A.: High-resolution grids of hourly meteorological variables for Germany, Theoretical and Applied Climatology, 131, 899–926, https://doi.org/10.1007/s00704-016-2003-7, 2018.

Hollis, D., McCarthy, M., Kendon, M., Legg, T., and Simpson, I.: HadUK-Grid—A new UK dataset of gridded climate observations, Geoscience Data Journal, 6, 151–159. https://doi.org/10.1002/gdj3.78, 2019.

Škrk, N., Serrano-Notivoli, R., Čufar, K., Merela, M., Črepinšek, Z., Kajfež Bogataj, L., and de Luis, M.: SLOCLIM: a high-resolution daily gridded precipitation and temperature dataset for Slovenia, Earth System Science Data, 13, 3577–3592. https://doi.org/10.5194/essd-13-3577-2021, 2021.

Doblas-Reyes, F. J., Sorensson, A. A., Almazroui, M., Dosio, A., Gutowski, W. J., Haarsma, R., Hamdi, R., Hewitson, B., Kwon, W.-T., Lamptey, B. L., Maraun, D., Stephenson, T. S., Takayabu, I., Terray, L., Turner, A., and Zuo, Z.: Linking global to regional climate change. In: Masson-Delmotte, V., Zhai, P., Pirani, A., Connors, S. L., Pean, C., Berger, S., Caud, N., Chen, Y., Goldfarb,

L., Gomis, M. I., Huang, M., Leitzell, K., Lonnoy, E., Matthews, J. B. R., Maycock, T. K., Waterfield, T., Yelekci, O., Yu, R., and Zhou, B. (eds.): Climate Change 2021: The Physical Science Basis. Contribution of Working Group I to the Sixth Assessment Report of the Intergovernmental Panel on Climate Change. Cambridge University Press, 2021.

Serrano-Notivoli, R., and Tejedor, E.: From rain to data: A review of the creation of monthly and daily station-based gridded precipitation datasets, Wiley Interdisciplinary Reviews: Water, 8, e1555, https://doi.org/10.1002/wat2.1555, 2021.

Pantillon, F., Davolio, S., Avolio, E., Calvo-Sancho, C., Carrió, D. S., Dafis, S., Gentile, E. S., Gonzalez-Aleman, J. J., Gray, S., Miglietta, M. M., Patlakas, P., Pytharoulis, I., Ricard, D., Ricchi, A., Sanchez, C., and Flaounas, E.: The crucial representation of deep convection for the cyclogenesis of Medicane Ianos, Weather and Climate Dynamics, 5, 1187–1205, https://doi.org/10.5194/wcd-5-1187-2024, 2024.

Patlakas, P., Chaniotis, I., Hatzaki, M., Kouroutzoglou, J., and Flocas, H. A.: The eastern Mediterranean extreme snowfall of January 2022: synoptic analysis and impact of sea-surface temperature, Weather, 79, 25–33, https://doi.org/10.1002/wea.4397, 2024.

Politi, N., Vlachogiannis, D., Sfetsos, A., and Nastos, P. T.: High-resolution dynamical downscaling of ERA-Interim temperature and precipitation using WRF model for Greece, Climate Dynamics, 57, 799–825, https://doi.org/10.1007/s00382-021-05741-9, 2021.

Heinke, J., Müller, C., and Gerten, D.: Limitations of machine learning in a spatial context, EGU General Assembly 2023, Vienna, Austria, 24–28 Apr 2023, EGU23-16096, https://doi.org/10.5194/egusphere-egu23-16096, 2023